JCB Journal of Cell Biology

# Vacuoles provide the source membrane for TORC1-containing signaling endosomes

Kenji Muneshige[1] and Riko Hatakeyama[1]

**Organelle biogenesis is fundamental to eukaryotic cell biology. Yeast signaling endosomes were recently identified as a signaling platform for the evolutionarily conserved Target of Rapamycin Complex 1 (TORC1) kinase complex. Despite the importance of signaling endosomes for TORC1-mediated control of cellular metabolism, how this organelle is generated has been a mystery. Here, we developed a system to induce synchronized *de novo* formation of signaling endosomes, enabling real-time monitoring of their biogenesis. Using this system, we identify vacuoles as a membrane source for newly formed signaling endosomes. Membrane supply from vacuoles is mediated by the CROP membrane-cutting complex, consisting of Atg18 PROPPIN and retromer subunits. The formation of signaling endosomes requires TORC1 activity, suggestive of a tightly regulated process. This study unveiled the first mechanistic principles and molecular participants of signaling endosome biogenesis.**

## Introduction

Eukaryotic cells are compartmentalized by membrane structures, or organelles, that perform specialized functions. Organelle biogenesis is fundamental for establishing cellular architecture and functions. Here, we examine the biogenesis of yeast signaling endosomes, a novel subpopulation of endosomes (or a novel organelle; see Discussion) we recently reported (Hatakeyama and De Virgilio, 2019a).

Signaling endosomes play critical roles in the regulation of cellular metabolism. Signaling endosomes accommodate on their surface the Target of Rapamycin Complex 1 (TORC1) kinase complex, an evolutionarily conserved master regulator of cell growth and metabolism (Shimobayashi and Hall, 2014; Liu and Sabatini, 2020). Being activated and inactivated by pro- and antigrowth stimulations, respectively, TORC1 regulates various cellular processes by phosphorylating substrate proteins. In yeast, TORC1 localizes to the surface of two organelles: vacuoles (the yeast counterpart of lysosomes) and signaling endosomes (Sturgill et al., 2008; Hatakeyama et al., 2019). Because TORC1 can only physically interact with and phosphorylate nearby proteins, the difference in its localization translates into a difference in its target substrates, and so the downstream biological processes controlled. We have so far reported three specific substrates of the signaling endosome-residing TORC1 pool (Hatakeyama et al., 2019; Hatakeyama and De Virgilio, 2019b; Chen et al., 2021): Atg13, a regulator of macroautophagy (Kamada et al., 2010); Vps27, a subunit of ESCRT-0 complex that regulates microautophagy (Oku et al.,

2017); and Fab1, a lipid kinase that generates phosphatidyl-inositol 3,5-bisphosphate (PI(3,5)P$_2$) from phosphatidylinositol 3-phosphate (PI3P) (Hasegawa et al., 2017). The vacuolar TORC1 pool, on the other hand, phosphorylates Sch9, a regulator of protein synthesis (Hatakeyama et al., 2019; Caligaris et al., 2023). Such division of labor between spatially distinct pools of TORC1 explains, at least in part, why TORC1 can regulate diverse biological processes operating at different subcellular locations.

The principle of spatially and functionally distinct TORC1 pools seems to be evolutionarily conserved as functional differences between lysosomal and cytoplasmic TORC1 pools were recently reported for mammalian cells (Fernandes et al., 2024). To date, however, signaling endosomes have only been described in budding yeast. Of note, the term "signaling endosome" also occurs in different contexts, either as a general concept or for other signaling pathways (Platta and Stenmark, 2011). These are distinct from the TORC1-containing signaling endosomes investigated in this study.

Apart from TORC1, signaling endosomes accommodate specific proteins and lipids that overlap with residents of vacuoles and/or canonical, TORC1-negative endosomes. The proteins shared by signaling endosomes and vacuoles include the EGO complex and Pib2, upstream regulators of TORC1 (Hatakeyama et al., 2019; Nicastro et al., 2017; Hatakeyama, 2021); Fab1, aforementioned lipid kinase (Chen et al., 2021); Ypt7, a yeast Rab7 small GTPase (Füllbrunn et al., 2024); Ivy1, a phospholipid-binding protein that regulates TORC1 and Fab1 (Gao et al., 2022); and

[1]Institute of Medical Sciences, School of Medicine, Medical Sciences and Nutrition, University of Aberdeen, Aberdeen, UK.

Correspondence to Riko Hatakeyama: riko.hatakeyama@abdn.ac.uk.

Yck3, a casein kinase (Grziwa et al., 2023). Signaling endosomes also accommodate proteins typically found on endosomes but not vacuoles such as Vps21, a yeast Rab5 small GTPase (Hatakeyama et al., 2019); Vps8, a subunit of the CORVET vesicle tethering complex (Chen et al., 2021); and aforementioned Vps27 (Hatakeyama et al., 2019). The phosphoinositide composition of the signaling endosomal membrane, i.e., enrichment of PI(3,5)P$_2$ and PI3P, is characteristic of both signaling endosomes and the vacuolar membrane (Chen et al., 2021). The "vacuole–endosome hybrid" composition of the signaling endosomal membrane suggests its close relation to both vacuoles and canonical endosomes. Moreover, signaling endosomes are always found adjacent to vacuoles. It is however unclear whether and how materials are exchanged between signaling endosomes and vacuoles or between signaling endosomes and canonical endosomes.

Curiously, not all cells in a population have signaling endosomes. Depending on the yeast strain background and exact growth conditions, typically, less than half of cells harbor detectable signaling endosomes at a time. Those that do usually have only one, sometimes two, signaling endosomes. These observations suggest that signaling endosomes are dynamically generated and turned over in a tightly regulated manner rather than being a long-lived static structure. Such dynamics of signaling endosomes are likely to have significant consequences for cellular metabolism, such as autophagic activity regulated by TORC1 signaling. Currently, however, the molecular mechanisms and the upstream stimuli/triggers of signaling endosome formation are unknown. Nor do we know where signaling endosomes come from, i.e., their membrane source.

In the present work, we addressed these fundamental questions by developing a system to induce and synchronize the formation of signaling endosomes, enabling real-time observation of their biogenesis. We obtained evidence that newly formed signaling endosomes' source membrane is from adjacent vacuoles. Mechanistically, this process requires the Atg18 protein and is supported by TORC1 activity. Our work thus provides a first clue for understanding signaling endosome biogenesis.

## Results

### Atg18 is required for signaling endosome formation

Studying the biogenesis of signaling endosomes in a wild-type cell population is challenging as we cannot predict in which cell biogenesis will happen and when. To overcome this difficulty, we designed a strategy to induce and synchronize the *de novo* formation of signaling endosomes. Our strategy involved two steps (Fig. 1 A). Our first step was to identify a gene, "Gene X," that is required for signaling endosome formation. The *Gene X∆* strain should have no signaling endosomes. In the second step, Gene X expression is placed under the control of a conditionally inducible promotor. In such a yeast strain, without Gene X induction, cells should behave as *Gene X∆* and thus should not have signaling endosomes. Once the expression of Gene X is induced, cells should start generating signaling endosomes, eventually behaving like wild-type cells. Time-lapse imaging during Gene X induction should enable real-time monitoring of *de novo* signaling endosome formation within a cell population.

A candidate Gene X was *ATG18*, which encodes a protein of the PROPPIN (β-propellers that bind polyphosphoinositides) family (Michell et al., 2006). We previously reported that *atg18∆* cells lose the perivacuolar, punctual localization of the PI(3,5)P$_2$ probe GFP-Sch9$^{1-183}$ that represents signaling endosomes (Chen et al., 2021). This observation suggested that *atg18∆* cells may lack signaling endosomes. We tested this possibility by visualizing Tor1, a catalytic subunit of TORC1 that defines signaling endosomes, by fusing it with mNeonGreen. We confirmed that mNeonGreen-tagged Tor1 is functional (Fig. S1, A and B). We found that *atg18∆* cells lack perivacuolar puncta of mNeonGreen-Tor1 (Fig. 1 B and Fig. S2 A). For other signaling endosome marker proteins, Gtr1 (an EGO complex subunit), Pib2, and Sch9$^{1-183}$, puncta indicative of signaling endosomes were similarly absent in *atg18∆* cells (Fig. 1, C–E and Fig. S2, B–D). The loss of puncta was not due to reduced expression of marker proteins because their vacuolar localization was intact. Note that the vacuolar enlargement observed in Fig. 1 is a known phenotype of *atg18∆* (Dove et al., 2004).

Ivy1 has been regarded as another marker of signaling endosomes. However, we noticed that Ivy1 puncta are significantly more abundant than the puncta of Tor1, Gtr1, Pib2, and Sch9$^{1-183}$. This observation suggests that Ivy1 localizes to other subcellular loci as well. Consistently, the Ivy1 puncta were reduced but not absent from *atg18∆* cells (Fig. 1 F). Most Ivy1 puncta were positive for Vps21 (Fig. S2, E and F), in agreement with a previous study (Gao et al., 2022), suggesting that they are endosomes (Fig. S2 G; see also Discussion). Ivy1 is thus not a specific marker of signaling endosomes.

To confirm that Atg18 is not required for the formation of other endosomal subpopulations, we examined the localization of Vps4 and Vps21, endosomal markers not specific to signaling endosomes (Hatakeyama et al., 2019; Chen et al., 2021). These proteins indeed retained their punctual localization pattern in *atg18∆* cells (Fig. 1, G and H), suggesting that the requirement of Atg18 is specific to signaling endosomes. We also confirmed that the trans-Golgi network marked by Sec7 is unaffected (Fig. 1 I).

Consistent with the direct involvement of Atg18 in signaling endosome formation and/or function, mCherry-Atg18 formed perivacuolar puncta partially overlapping with Tor1 puncta (Fig. 1 J). The existence of Atg18-negative Tor1 puncta suggests that Atg18 may localize to signaling endosomes only transiently. Of note, our mCherry-Atg18 was functional for signaling endosome formation, vacuolar fission, and autophagy as shown in Fig. S1, C–E. Collectively, our observations suggest an essential role for Atg18 in signaling endosome formation.

### The Atg18-containing CROP complex mediates signaling endosome biogenesis

Having established the requirement of Atg18 for signaling endosome formation, we addressed the underlying molecular mechanisms. We first tested whether this function of Atg18 is shared by other structurally similar PROPPINs Atg21 and Hsv2 (Krick et al., 2008). Signaling endosomes were observed at normal levels in *atg21∆* and *hsv2∆* cells (Fig. 2 A; and Fig. S3, A

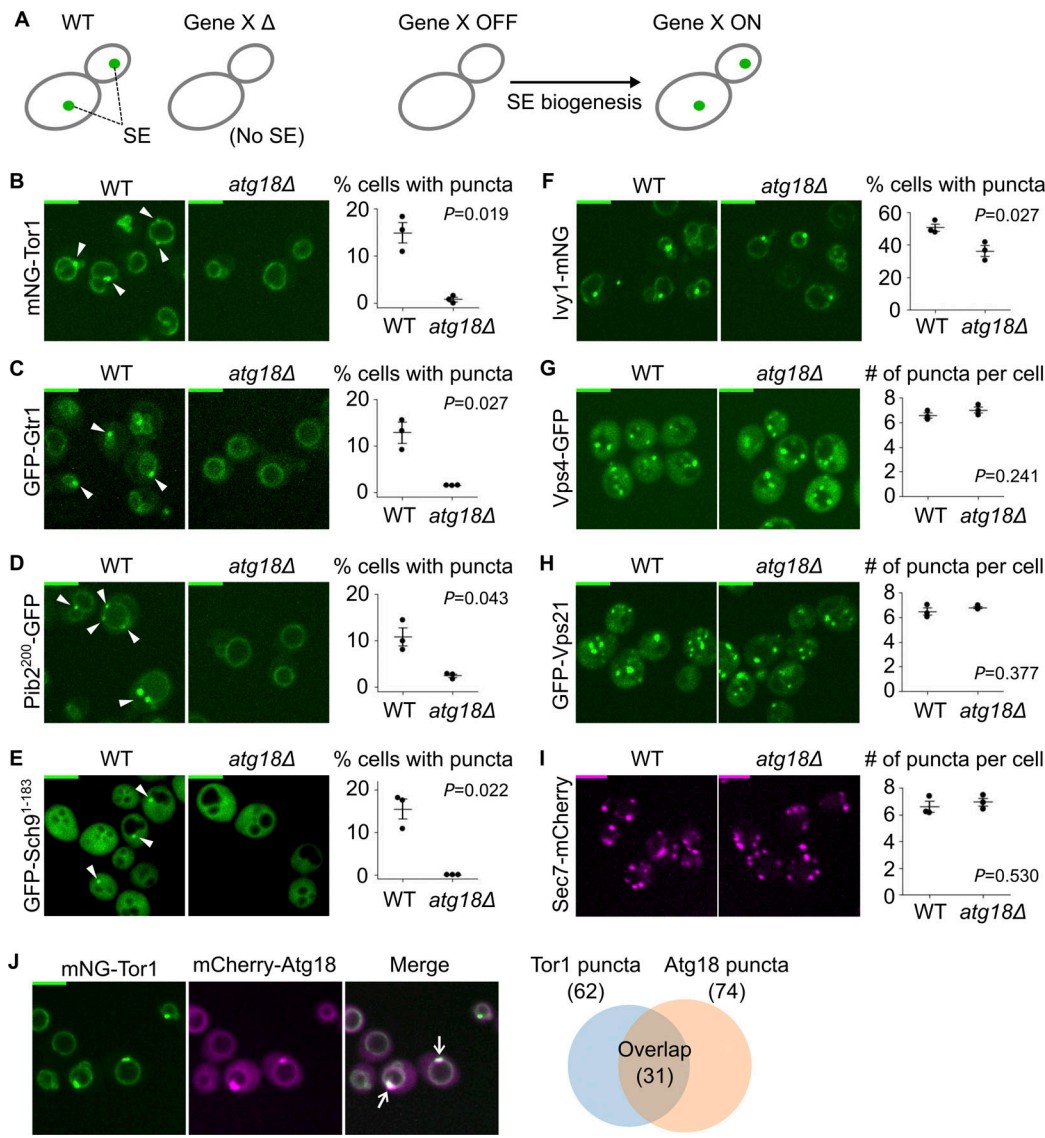

Figure 1. **Atg18 is required for signaling endosome formation. (A)** Strategy to induce and synchronize *de novo* formation of signaling endosomes (SE). See text for details. **(B–I)** Requirement of Atg18 for the formation of signaling endosomes but not that of canonical endosomes or trans-Golgi network. Wild-type (WT) and *atg18Δ* cells expressing the indicated fusion proteins were analyzed by fluorescent microscopy. Arrowheads indicate signaling endosomes. Quantifications of puncta-containing cells are shown on the right (mean ± SEM, >100 cells were analyzed for three independent biological replicates). In G–I, because all the cells contained puncta, the number of puncta per cell is presented. **(J)** Localization of Atg18 to signaling endosomes. Cells co-expressing mNeonGreen-Tor1 (mNG-Tor1) and mCherry-Atg18 were analyzed by fluorescent microscopy. Arrows indicate colocalization of Tor1 and Atg18 at signaling endosomes. Quantification of colocalization is shown in the Venn diagram on the right. Scale bars: 5 μm.

and B), suggesting that this function is unique to Atg18 among the three yeast PROPPINs.

Atg18 functions in the context of distinct protein complexes (Fig. 2 B). The Atg18–Atg2–Atg9 trimeric complex promotes the expansion of the preautophagosomal membrane by transferring lipid molecules from the endoplasmic reticulum (Maeda et al., 2019; Valverde et al., 2019; Gómez-Sánchez et al., 2018). More recently, the Mayer and Thumm groups independently reported another Atg18-containing complex, the CROP (Cutting Retromer-on-PROPPIN) complex (Courtellemont et al., 2022; Marquardt et al., 2023; Gopaldass and Mayer, 2024; Marquardt and Thumm, 2023). CROP is comprised of Atg18, Vps26, Vps29, and Vps35, the latter three of which are subunits of the retromer

sorting complex (Seaman et al., 1998). CROP cuts the membrane of endo-lysosomal compartments, promoting the formation of tubulo-vesicular transport carriers from endosomes, and vacuolar fission and fragmentation (De Leo et al., 2021; Gopaldass et al., 2017; Courtellemont et al., 2022; Marquardt et al., 2023).

We addressed the protein complex and functions through which Atg18 supports signaling endosome formation by deleting each binding partner. Signaling endosomes were intact in *atg2Δ* and *atg9Δ* cells (Fig. 2 C), ruling out the involvement of the Atg18–Atg2–Atg9 complex. In contrast, *vps26Δ*, *vps29Δ*, and *vps35Δ* mutants had significantly fewer signaling endosomes than wild-type cells (Fig. 2 D; and Fig. S3, C and D). Because the phenotype of retromer mutants can be caused by either the loss

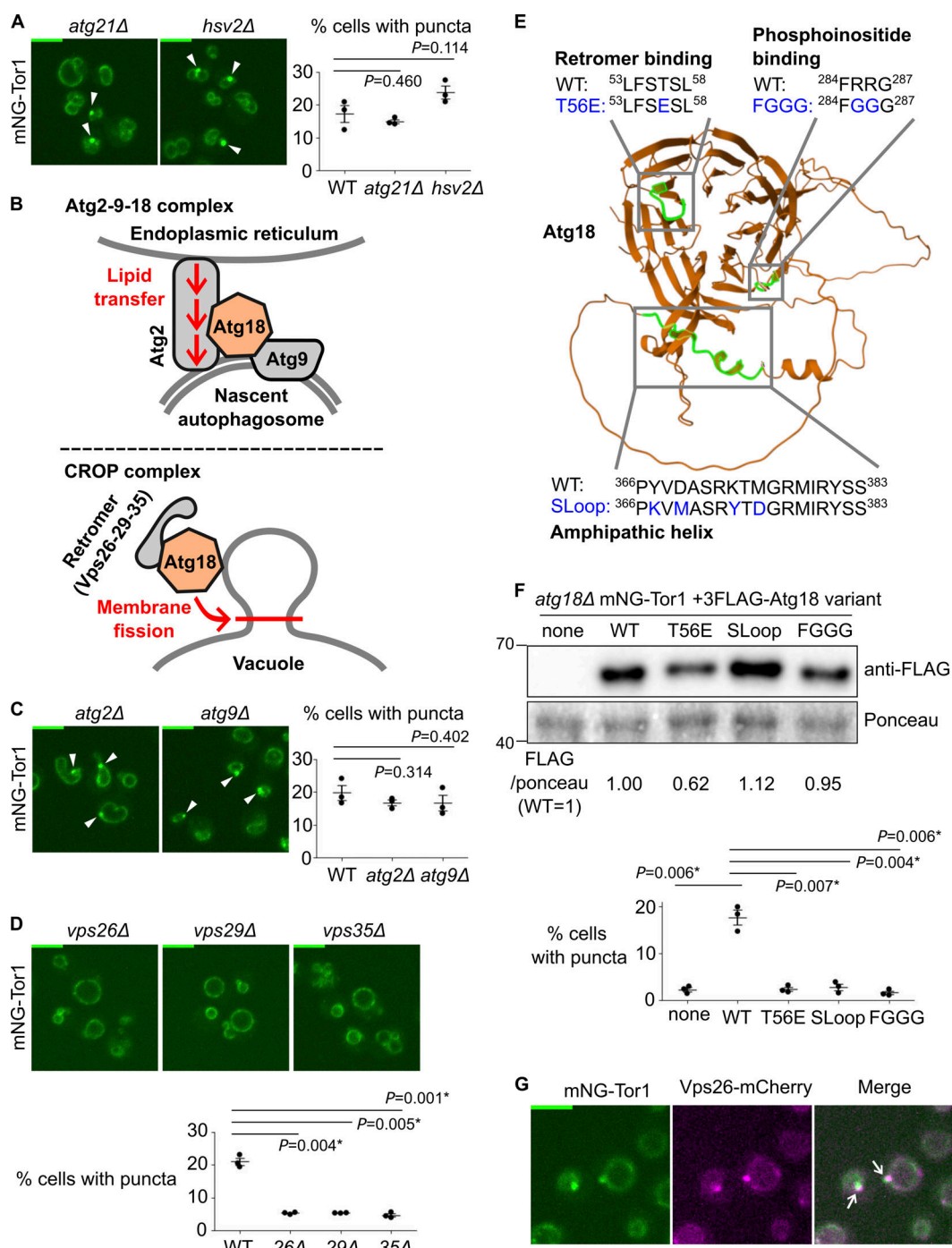

Figure 2. **Atg18-containing CROP complex is required for signaling endosome formation. (A)** No requirement for other PROPPIN proteins for signaling endosome formation. The indicated strains expressing mNeonGreen-Tor1 (mNG-Tor1) were analyzed by fluorescent microscopy. Arrowheads indicate signaling endosomes. Quantifications of puncta-containing cells are shown on the right. **(B)** Compositions and functions of Atg18-containing protein complexes. See text for details. **(C)** No requirement for the Atg2-9-18 complex for signaling endosome formation. The indicated strains expressing mNG-Tor1 were analyzed by fluorescent microscopy. Arrowheads indicate signaling endosomes. Quantifications of puncta-containing cells are shown on the right. **(D)** Requirement of the CROP complex subunits for signaling endosome formation. The indicated strains expressing mNG-Tor1 were analyzed by fluorescent microscopy. Quantifications of puncta-containing cells are shown at the bottom. **(E)** Atg18 domains required for the formation and/or membrane-cutting activity of the CROP complex. The CROP-relevant domains are highlighted in green on the AlphaFold-predicted structure of Atg18. Loss-of-function mutations examined in this study are indicated in blue. **(F)** Requirement of CROP-relevant Atg18 domains for signaling endosome formation. atg18Δ cells expressing mNG-Tor1 and the indicated 3FLAG-Atg18 variants were analyzed by western blotting (top) and fluorescent microscopy. Quantifications of puncta-containing cells are shown at the bottom. **(G)** Localization of Vps26 to signaling endosomes. Cells co-expressing mNG-Tor1 and Vps26-mCherry were analyzed by fluorescent microscopy. Arrows indicate colocalization of Tor1 and Vps26 at signaling endosomes. Error bars represent mean ± SEM. More than 100 cells were analyzed for three independent biological replicates. Scale bars: 5 μm. The numbers next to blots denote molecular weights (kDa). Source data are available for this figure: SourceData F2.

of the CROP complex or that of other retromer-dependent trafficking events, we next examined Atg18 mutants defective in the CROP function. We analyzed three Atg18 mutants specifically defective in CROP complex formation and/or membrane fission: T56E, SLoop, and FGGG mutants (Fig. 2 E) (De Leo et al., 2021; Courtellemont et al., 2022; Gopaldass and Mayer, 2024). As expected, these Atg18 mutants (which were all expressed robustly though at varied levels; Fig. 2 F, top) failed to support signaling endosome formation (Fig. 2 F, bottom). The 3FLAG–Atg18 construct used here was functional for signaling endosome formation, vacuolar fission, and autophagy as shown in Fig. S1, F–I.

Our results suggest a major role for the CROP complex in signaling endosome formation. Consistently, retromer subunits localized to signaling endosomes along with other endosomal subpopulations (Fig. 2 G and Fig. S3, E–G). We nevertheless noticed that the defects in retromer mutants (Fig. 2 D) are significant but only partial, differing from the complete loss of signaling endosomes in atg18Δ cells. This observation aligns well with the fact that Atg18 can act alone to cut a membrane, albeit less efficiently than when it is in the CROP complex (Gopaldass et al., 2017; Courtellemont et al., 2022).

### Development of an inducible system to study signaling endosome biogenesis

The participation of the CROP complex indicates that signaling endosome biogenesis involves a membrane fission process. Because (1) the CROP complex cuts the vacuolar membrane, (2) signaling endosomes are always adjacent to vacuoles, and (3) the protein and lipid composition significantly overlap between signaling endosomes and vacuoles, it was plausible that vacuoles supply the membrane, through a fission process, to newly born signaling endosomes.

We sought to test this hypothesis by monitoring signaling endosome formation in real-time. To this end, we took advantage of the complete loss of signaling endosomes in atg18Δ cells, identifying ATG18 as Gene X (Fig. 1 A). To conditionally express Atg18, we utilized the WTC$_{846}$ toolkit, an improved tetracycline-inducible gene expression system (Azizoğlu et al., 2021). We confirmed that anhydrotetracycline (aTc) induces 3FLAG–Atg18 expression in a dose-dependent manner in the WTC$_{846}$-ATG18 strain (Fig. 3 A and Fig. S4 A). Without aTc treatment, Atg18 expression was undetectable. The Atg18 expression level with 1 ng/ml aTc approximately corresponded to the endogenous level (Fig. 3 A first lane), so we used this concentration of aTc in the following experiments unless specified otherwise. The expression level of Tor1 did not change after aTc treatment, meaning that aTc-dependent changes in Tor1 distribution described below are not due to changes in the Tor1 abundance.

As expected, WTC$_{846}$-ATG18 cells showed no Tor1 puncta without aTc treatment. Tor1 puncta started to appear around 30–60 min after the addition of aTc (Fig. 3 B and Fig. S2 H). To confirm that these Tor1 puncta are bona fide signaling endosomes rather than fragmented vacuoles, we simultaneously visualized proteins that discriminate the two structures. We first examined the vacuolar ATPase subunit Vph1, which is present on vacuoles but not on signaling endosomes. Tor1 puncta barely

overlapped with Vph1, which did not form puncta upon Atg18 expression as expected (Fig. 3 C). A small fraction (5.8%) of Tor1 puncta was however positive for Vph1, which we interpret as representing rare vacuoles that are too small to be distinguishable from signaling endosomes. We then examined the Rab5 small GTPase Vps21, which is present on signaling endosomes but not on vacuoles. A fraction of Vps21 indeed colocalized with Tor1 puncta (Fig. 3 D), further confirming the identity of these Tor1 puncta as signaling endosomes.

Time-lapse imaging allowed us to track specific cells and monitor the formation of Tor1 puncta in real-time (Fig. 3 E; Airyscan higher-resolution images in Fig. 3 F). Hence, our WTC$_{846}$-ATG18 system enabled direct observation of signaling endosome biogenesis.

### The formation of signaling endosomes is linked to cell cycle progression

Curiously, not all cells formed Tor1 puncta even when Atg18 was overexpressed (Fig. 3 B; 10 ng/ml aTc). This heterogeneity among a cell population cannot solely be attributed to variations in the expression nor puncta formation of Atg18 (Fig. S4 B), suggesting the existence of additional cue(s) required for signaling endosome formation.

In search of additional factors for signaling endosome formation, we examined the link to cellular morphology, specifically, the budding status. Interestingly, in small-budded cells, daughter cells rarely formed signaling endosomes after Atg18 induction (Fig. S4 C). Daughter cells seem to acquire the ability to form signaling endosomes as the cell cycle progresses (i.e., in large-budded cells). A similar trend was observed in wild-type, steady-state cells. The number of signaling endosomes, which should reflect the balance of their formation and degradation in this case, was low in daughters in small-budded cells (Fig. S4 D). These observations suggest that the cell cycle progression impacts the ability of cells to form signaling endosomes.

### Signaling endosomes inherit the vacuolar membrane

Using the WTC$_{846}$-ATG18 system, we tested the vacuole-origin hypothesis by examining the vacuolar membrane during de novo formation of signaling endosomes. As a first approach, we tracked vacuolar lipids using the lipophilic dye FM4-64, an established marker of the vacuolar membrane. FM4-64, once added to the culture media, is initially inserted into the plasma membrane and then unidirectionally trafficked to vacuoles through endocytosis (Vida and Emr, 1995). When FM4-64 is washed out shortly after its addition, and allowed to complete its endocytic trafficking (which takes 30–60 min), it specifically stains the vacuolar membrane. We stained the vacuolar membrane of WTC$_{846}$-ATG18 cells with FM4-64 prior to Atg18 induction (Fig. 4 A, time 0). Because FM4-64 has been washed out, any FM4-64 signal observed afterward must have come from the vacuolar membrane. Around 30–60 min after Atg18 induction, we started observing the formation of perivacuolar, FM4-64–positive puncta. These puncta were positive for Tor1, identifying them as signaling endosomes. This observation supports the idea that newly formed signaling endosomes acquire lipids from vacuoles.

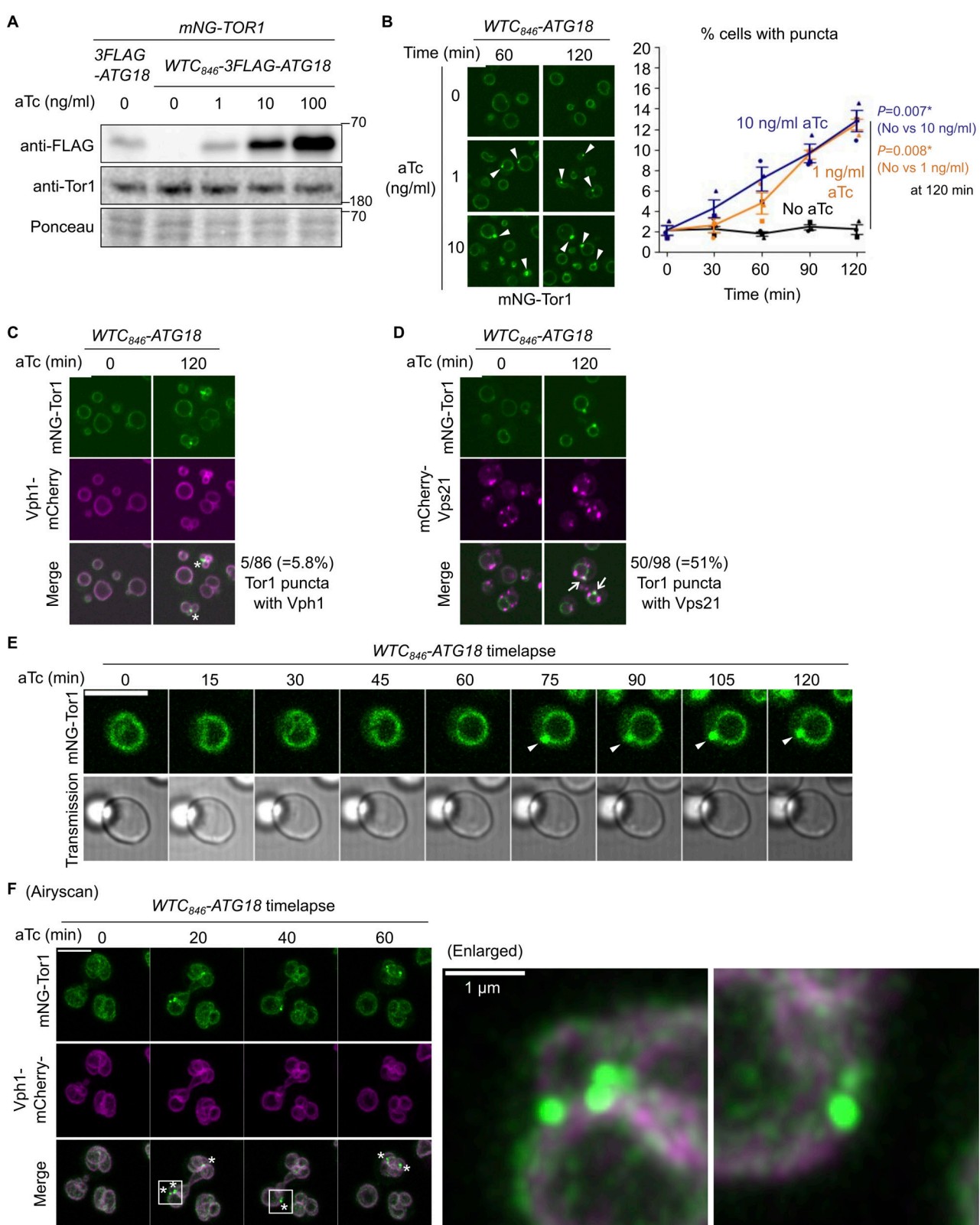

**Figure 3.  Development of a signaling endosome induction system. (A)** Confirmation of WTC$_{846}$-induced Atg18 expression. The indicated strains were treated with aTc at the indicated final concentrations for 60 min and analyzed by western blotting. **(B)** WTC$_{846}$-induced formation of signaling endosomes. *WTC$_{846}$-ATG18* cells expressing mNeonGreen-Tor1 (mNG-Tor1) were treated with aTc at the indicated final concentrations for the indicated periods and analyzed by fluorescent microscopy. Arrowheads indicate newly formed signaling endosomes. Quantifications of puncta-containing cells are shown on the right (mean ± SEM, >100 cells were analyzed for three biological replicates). **(C)** Absence of v-ATPase on induced signaling endosomes. *WTC$_{846}$-ATG18* cells co-expressing mNG-Tor1 and Vph1-mCherry were treated with 1 ng/ml aTc for the indicated periods and analyzed by fluorescent microscopy. Asterisks indicate

the absence of Vph1 puncta on newly formed signaling endosomes. **(D)** Localization of Vps21 to induced signaling endosomes. *WTC₈₄₆-ATG18* cells co-expressing mNG-Tor1 and mCherry-Vps21 were treated with 1 ng/ml aTc for the indicated periods and analyzed by fluorescent microscopy. Arrows indicate colocalization of Tor1 and Vps21 at newly formed signaling endosomes. **(E)** Time-lapse monitoring of signaling endosome formation. *WTC₈₄₆-ATG18* cells expressing mNG-Tor1 were treated with 1 ng/ml aTc and analyzed by time-lapse fluorescent microscopy. Arrowheads indicate newly formed signaling endosomes. **(F)** Airyscan super-resolution time-lapse monitoring of signaling endosome formation. *WTC₈₄₆-ATG18* cells co-expressing mNG-Tor1 and Vph1-mCherry were treated with 10 ng/ml aTc and analyzed by time-lapse fluorescent microscopy. Asterisks indicate the absence of Vph1 puncta on newly formed signaling endosomes. Enlarged images of the region marked by the box are shown on the right. Scale bars: 5 μm except for the enlarged image in F. The numbers next to blots denote molecular weights (kDa). Source data are available for this figure: SourceData F3.

Unlike in the induced condition, signaling endosomes were not frequently stained with FM4-64 in a wild-type, steady-state cell population (Fig. 4 B). Because most signaling endosomes should have been pre-existing in this condition, we interpret this result as the lack of bulk lipid transport from vacuoles to mature signaling endosomes. Thus, the membrane of mature signaling endosomes seems to be no longer physically connected to the vacuolar membrane, as expected from the membrane scission activity of the CROP complex (see also Discussion). The relatively small fraction (30%) of FM4-64–positive signaling endosomes in a steady state might represent new/premature signaling endosomes and/or those retaining FM4-64 delivered through non-vacuolar routes (e.g., the endocytic route).

We then asked whether lipids are transferred as individual molecules, for example, via lipid transfer across a membrane contact site (Scorrano et al., 2019) or in the form of a membrane. The latter scenario would be supported if lipids accompany membrane proteins. None of the known signaling endosomal proteins however serves as a suitable marker because they are actually soluble proteins that are either peripherally attached or merely anchored to the membrane and can potentially transfer between organelles through cytoplasmic diffusion. We therefore searched for a membrane-spanning protein that localizes to both vacuoles and signaling endosomes. Searching a genome-wide protein localization database, YeastRGB (Dubreuil et al., 2019), identified the amino acid transporter Atg22 (Yang et al., 2006) as a membrane-spanning protein localizing to vacuoles and perivacuolar puncta. We confirmed these perivacuolar puncta are signaling endosomes by colocalizing them with Tor1 puncta (Fig. 5 A). Atg22 is therefore the first transmembrane protein that localizes to signaling endosomes, providing a useful marker for tracking the membrane.

We tracked Atg22 during Atg18 induction. Without induction, Atg22 localized solely to the vacuolar membrane as expected (Fig. 5 B). Upon induction, Atg22 migrated to newly formed signaling endosomes marked by Tor1. We obtained the same result even when the expression of Atg22 was shut off (using the conditional *GAL* promotor) prior to Atg18 induction (Fig. 5 C), suggesting the migration of pre-existing (rather than newly synthesized) Atg22 from vacuoles to signaling endosomes.

Together, our FM4-64– and Atg22-tracking experiments demonstrate that lipids and a membrane-integrated protein migrate from the vacuolar membrane during the formation of signaling endosomes. These results establish vacuoles as a membrane source of newly formed signaling endosomes.

**The migration of vacuolar proteins to signaling endosomes is a selective process**

It is worth noting that not all the vacuolar membrane proteins are transferred to signaling endosomes. Vph1, for example, remains confined to the vacuolar membrane as already mentioned (Fig. 3 C). To better understand which proteins are transferred to signaling endosomes, we set out to examine 16 additional vacuolar proteins for their localization at signaling endosomes. We visualized cells from GFP- or mNeonGreen-fusion strain libraries (Dubreuil et al., 2019) expressing mCherry-Gtr1 (which is functional; Fig. S1 J) as a marker for signaling endosomes.

Because Atg22 is an amino acid transporter, we examined other known members of vacuolar amino acid transporters (Kawano-Kawada et al., 2018). Among these transporters, we were able to recover seven GFP/mNeonGreen-tagged library strains showing the expected vacuolar localization. We found that Avt1, Avt3, and Avt6, which belong to the AVT family, localize to signaling endosomes (Fig. S5 A). The other four transporters, Vba1 and Vba4 from the VBA family, and Ypq1 and Ypq2 from the PQ-loop family, did not localize to signaling endosomes. Thus, as far as tested, localization of amino acid transporters to signaling endosomes is limited to the members of the AVT family and Atg22 (which forms a separate family on its own).

Atg22 is trafficked from the Golgi apparatus to vacuoles via the AP-3 vesicle trafficking pathway. We, therefore, examined eight additional, randomly chosen AP-3 cargoes (Eising et al., 2022). Among them, Pmc1, a vacuolar calcium transporter, and Bpt1, a vacuolar glutathione conjugate transporter, localized to signaling endosomes, whereas six others did not (Fig. S5 B). Hence, AP-3–mediated vacuolar delivery does not necessarily license cargo proteins for subsequent sorting to signaling endosomes.

We suspect that signaling endosomes mediate retrograde trafficking from the vacuolar membrane to canonical endosomes (see Discussion). We, therefore, asked if the recycling cargo Atg27 (Suzuki and Emr, 2018b) passes through signaling endosomes. Atg27 however did not localize to signaling endosomes (Fig. S5 C). This might be because the recycling route for Atg27, which is mediated by the Snx4 sorting nexin, is distinct from the PI(3,5)P₂/Atg18-mediated route (Suzuki and Emr, 2018a).

Along with vacuolar proteins, we examined the plasma membrane amino acid transporter Dip5 (Hatakeyama et al., 2010) because recent studies suggested that the endocytic pathway feeds into signaling endosomes (Gao et al., 2022; Füllbrunn et al., 2024). Indeed, we found Dip5 passing through signaling endosomes (Fig. S5 D). Thus, signaling endosomes are at the intersection of the endocytic pathway and a vacuole-derived membrane flow (Fig. S2 G; see also Discussion).

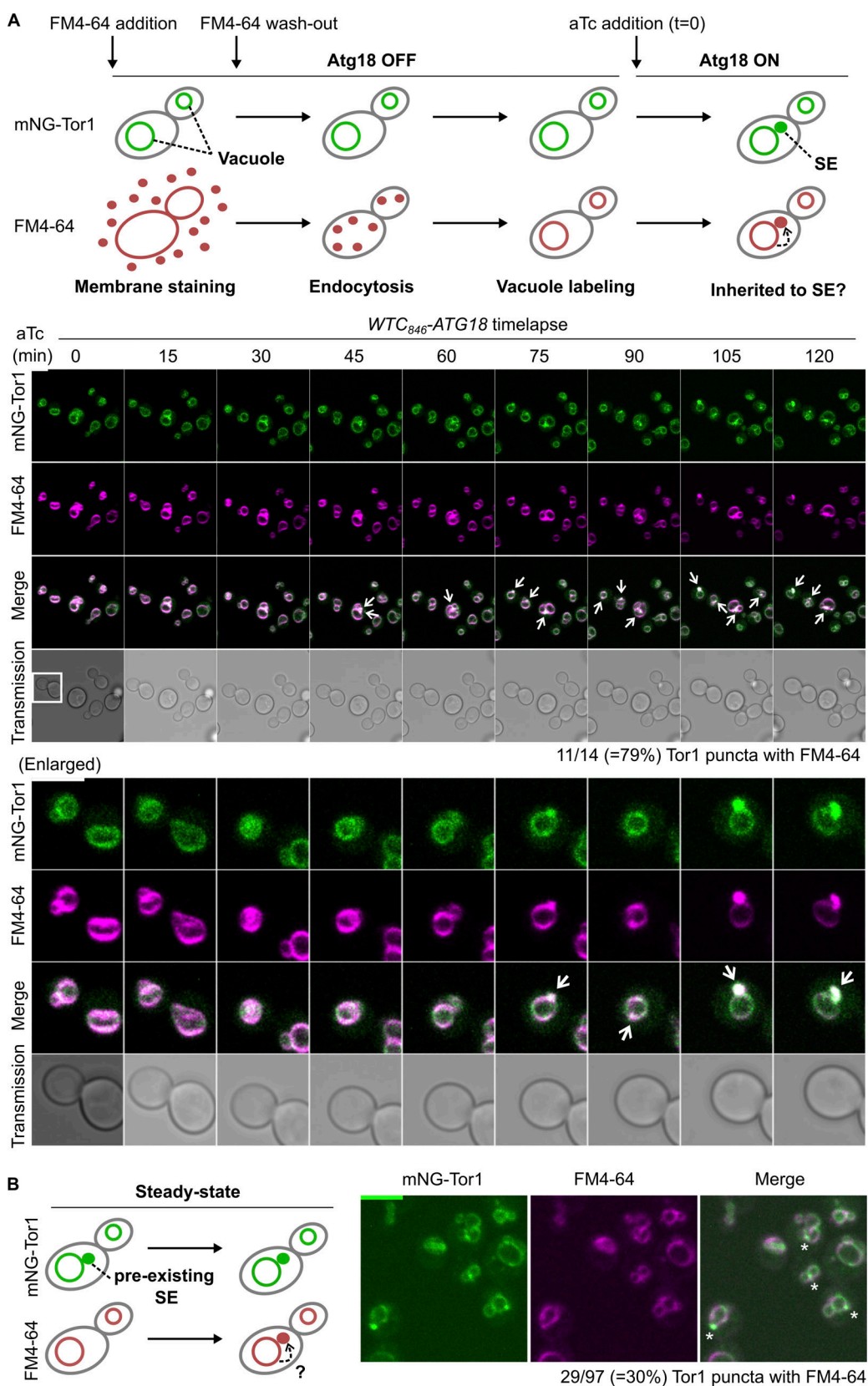

**Figure 4. Membrane lipid inheritance from vacuoles to newly formed signaling endosomes. (A)** Inheritance of vacuolar membrane lipids to newly formed signaling endosomes. Top: Experimental scheme. SE: signaling endosome. Middle: *WTC₈₄₆-ATG18* cells expressing mNeonGreen-Tor1 (mNG-Tor1) were prestained with FM4-64, treated with 1 ng/ml aTc, and analyzed by time-lapse fluorescent microscopy. Arrows indicate Tor1 localization and FM4-64 staining overlapping at newly formed signaling endosomes. Enlarged images of the cell marked by the box are shown at the bottom. **(B)** The limited FM4-64 staining of steady-state signaling endosomes. Left: Experimental scheme. Right: Wild-type cells expressing mNG-Tor1 were stained with FM4-64 and analyzed by fluorescent microscopy. Asterisks indicate the absence of FM4-64 signal on signaling endosomes. Scale bars: 5 µm.

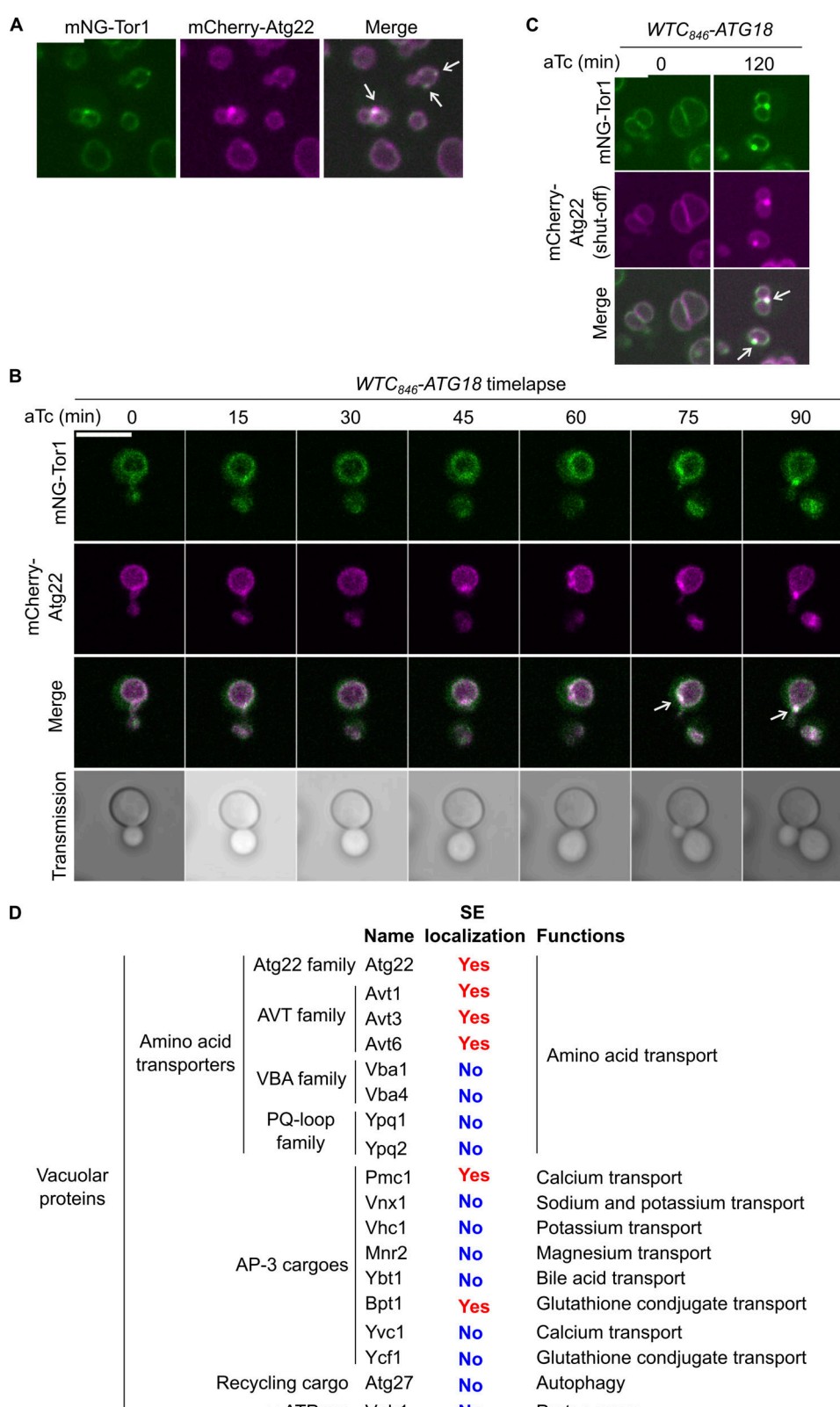

| | | Name | SE localization | Functions |
|---|---|---|---|---|
| Vacuolar proteins | Amino acid transporters — Atg22 family | Atg22 | Yes | Amino acid transport |
| | AVT family | Avt1 | Yes | |
| | | Avt3 | Yes | |
| | | Avt6 | Yes | |
| | VBA family | Vba1 | No | |
| | | Vba4 | No | |
| | PQ-loop family | Ypq1 | No | |
| | | Ypq2 | No | |
| | AP-3 cargoes | Pmc1 | Yes | Calcium transport |
| | | Vnx1 | No | Sodium and potassium transport |
| | | Vhc1 | No | Potassium transport |
| | | Mnr2 | No | Magnesium transport |
| | | Ybt1 | No | Bile acid transport |
| | | Bpt1 | Yes | Glutathione condjugate transport |
| | | Yvc1 | No | Calcium transport |
| | | Ycf1 | No | Glutathione condjugate transport |
| | Recycling cargo | Atg27 | No | Autophagy |
| | v-ATPase | Vph1 | No | Proton pump |

Figure 5. **Membrane protein inheritance from vacuoles to newly formed signaling endosomes. (A)** Localization of Atg22 to signaling endosomes. Cells co-expressing mNeonGreen-Tor1 (mNG-Tor1) and mCherry-Atg22 were analyzed by fluorescent microscopy. Arrows indicate colocalization of Tor1 and Atg22 at signaling endosomes (SE). **(B)** Inheritance of Atg22 from the vacuolar membrane to newly formed signaling endosomes. $WTC_{846}$-$ATG18$ cells co-expressing mNG-Tor1 and mCherry-Atg22 were treated with 1 ng/ml aTc and analyzed by time-lapse fluorescent microscopy. Arrows indicate colocalization of Tor1 and Atg22 at newly formed signaling endosomes. **(C)** Inheritance of pre-existing Atg22 from the vacuolar membrane to newly formed signaling endosomes. $WTC_{846}$-$ATG18$ cells co-expressing mNG-Tor1 (under the native promotor) and mCherry-Atg22 (under the inducible $GAL$ promotor) were cultured first in raffinose media

and then in galactose media to induce the expression of mCherry-Atg22. Cells were then cultured in glucose media to shut off the *de novo* expression of mCherry-Atg22, treated with 1 ng/ml aTc for the indicated periods, and analyzed by fluorescent microscopy. Arrows indicate colocalization of Tor1 and Atg22 at newly formed signaling endosomes. **(D)** Summary of the localization of vacuolar proteins to signaling endosomes. Strains co-expressing the indicated GFP- or mNeonGreen-fusion vacuolar proteins and mCherry-Gtr1 were analyzed by fluorescent microscopy. See Fig. S5, A–C for the actual images. Scale bars: 5 µm.

Fig. 5 D summarizes the localization of the tested vacuolar proteins to signaling endosomes. This list highlights the selective nature of protein transfer from the vacuolar membrane to signaling endosomes.

### TORC1-Fab1 signaling promotes signaling endosome biogenesis

We next asked for the signals that trigger the formation of signaling endosomes. Our Atg18 induction system enabled us to specifically test the effect of TORC1 activity. When TORC1 was inhibited by rapamycin, the Atg18-driven formation of Tor1 puncta was severely impaired (Fig. 6 A). We noticed a marginal impairment of Atg18 protein induction under rapamycin treatment (Fig. 6 B), as expected from TORC1's role in global protein translation. This reduction however does not explain the observed lack of signaling endosome formation because the Atg18 level induced in the presence of rapamycin was similar to its endogenous expression level (first lane).

Having established a requirement of TORC1 activity for signaling endosome formation, we searched for the relevant phosphorylation substrate. One TORC1 substrate functionally linked to Atg18 is the Fab1 lipid kinase, which produces PI(3,5)P$_2$ that binds to Atg18 and promotes its membrane-fission activity (Gopaldass et al., 2017). TORC1 directly phosphorylates Fab1, stimulating its relocation from vacuoles to signaling endosomes (Chen et al., 2021). This triggers the local production of PI(3,5)P$_2$ at signaling endosomes, which in turn anchors TORC1 to the membrane, serving as a positive feedback mechanism to maintain functional signaling endosomes. It has been unknown whether Fab1 phosphorylation also plays a role in the initial formation of signaling endosomes.

We examined the effect of Fab1 phosphorylation using its phospho-deficient (*fab1-6A*) and -mimetic (*fab1-6D*) mutants (Chen et al., 2021). Upon Atg18 induction, *fab1-6A* formed fewer signaling endosomes compared with wild-type cells (Fig. 6 C). Conversely, *fab1-6D* mutants formed more signaling endosomes (Fig. 6 C). These observations suggest that Fab1 phosphorylation indeed promotes signaling endosome biogenesis. Interestingly, the signal area of Tor1 puncta was bigger in *fab1-6D* cells (Fig. S2 I), suggesting a larger size of signaling endosomes and/or an excessive accumulation of TORC1 in these cells. Rapamycin exerted its inhibitory effect even in *fab1-6D* cells, suggesting the existence of additional TORC1 substrate(s), and/or additional TORC1-phosphorylation site(s) on Fab1 that are required for signaling endosome formation.

Our results suggest a positive regulatory role for the TORC1-Fab1 signaling in *de novo* formation of signaling endosomes, presumably through the recruitment and/or activation of Atg18. Given that Atg18 regulates Fab1 (Efe et al., 2007; Jin et al., 2008), these results implicate Atg18 with the TORC1-Fab1 module in forming an extended, tightly interconnected regulatory feedback loop (Fig. 6 D).

## Discussion

The TORC1-positive perivacuolar structure has been designated the signaling endosome because it accommodates proteins typically found on endosomes (Hatakeyama and De Virgilio, 2019a). In the present study, we identify vacuoles as a membrane source for newly formed signaling endosomes. A bulk membrane flow from vacuoles at the same scale (i.e., being visible with FM4-64) does not happen to canonical endosomes, for which the membrane is supplied mainly from the plasma membrane and the Golgi apparatus via intracellular vesicle trafficking. This unique mode of biogenesis suggests that signaling endosomes should be most correctly viewed as a distinct organelle rather than a mere subpopulation of canonical endosomes. We note, however, that our vacuole-origin model does not rule out membrane supply from other organelles, including canonical endosomes, to signaling endosomes (see below for more discussion).

At the mechanistic level, the requirement for Atg18, particularly in the context of the CROP complex, strongly suggests that signaling endosomes are formed through fission of the vacuolar membrane (Fig. 6 D). This process would be topologically similar to the vacuolar fragmentation process, which is also mediated by the CROP complex. The two processes are however distinct in terms of symmetry. Vacuolar fragmentation is a symmetric process via which a parental vacuole is divided into qualitatively identical daughter vacuoles. In contrast, signaling endosomes differ from vacuoles in terms of composition and function. This fundamental difference between the two processes suggests the existence of molecular machinery uniquely involved in signaling endosome biogenesis.

The scarcity of FM4-64–stained signaling endosomes in a steady state (Fig. 4 B) favors the idea that the membrane of mature signaling endosomes is discontinuous from the vacuolar membrane. Because signaling endosomes are always found adjacent to the vacuolar membrane, the two organelles may form membrane contact sites. Nevertheless, the direct visualization of the interface between signaling endosomes and vacuoles is currently missing. Further analyses with higher resolution techniques (e.g., electron microscopy) are needed to clarify the physical relationship between signaling endosomes and vacuoles.

Questions remain concerning how the membrane of signaling endosomes is differentiated from the vacuolar membrane. There must exist a selection mechanism that targets certain vacuolar proteins (such as Atg22) to signaling endosomes while retaining others (such as v-ATPase) on the vacuolar membrane (Fig. 5 D). One possibility is that Atg22, etc., contains a signaling endosome-targeting motif, and/or that v-ATPase, etc., has a

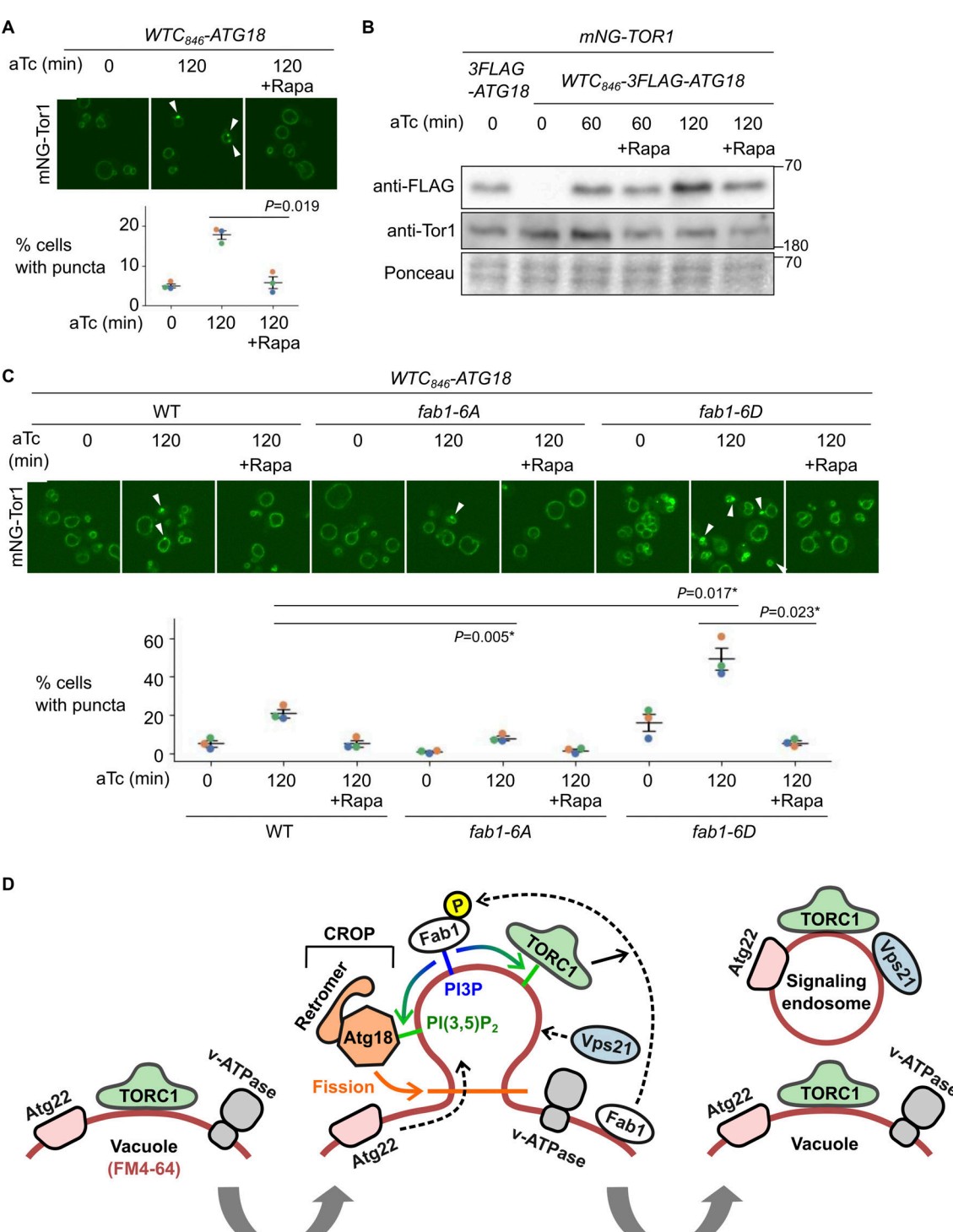

Figure 6.  **TORC1-Fab1 signaling promotes signaling endosome biogenesis. (A)** Requirement of TORC1 activity for signaling endosome formation. $WTC_{846}$-*ATG18* strains expressing mNeonGreen-Tor1 (mNG-Tor1) were treated with 1 ng/ml aTc for the indicated periods in the presence or absence of 200 ng/ml rapamycin (Rapa) and analyzed by fluorescent microscopy. Arrowheads indicate newly formed signaling endosomes. Quantifications of puncta-containing cells are shown at the bottom. **(B)** Confirmation of Atg18 induction in the presence of rapamycin. The indicated strains were treated with 1 ng/ml aTc for the indicated periods in the presence or absence of 200 ng/ml rapamycin and analyzed by western blotting. **(C)** Fab1 phosphorylation by TORC1 promotes signaling endosome formation. Wild-type, *fab1-6A*, or *fab1-6D* cells expressing mNG-Tor1 were treated with 1 ng/ml aTc for the indicated periods in the presence or absence of 200 ng/ml rapamycin and analyzed by fluorescent microscopy. Arrowheads indicate newly formed signaling endosomes. Quantifications of puncta-containing cells are shown at the bottom. **(D)** Model of signaling endosome biogenesis. Fab1 phosphorylation by TORC1 triggers its relocation to the site of signaling endosome formation on the vacuolar membrane. There, Fab1 locally produces $PI(3,5)P_2$ from PI3P. $PI(3,5)P_2$ anchors and stabilizes TORC1, which in turn further phosphorylates and stabilizes Fab1, resulting in a positive feedback loop. In the meantime, $PI(3,5)P_2$ recruits and activates the Atg18-containing CROP complex, which cuts the vacuolar membrane and thereby separates signaling endosomes. During this process, vacuolar membrane lipids (labeled by

FM4-64) as well as certain membrane-spanning proteins such as Atg22 migrate into forming signaling endosomes. Other membrane-spanning proteins such as v-ATPase remain on the vacuolar membrane. Certain endosomal proteins such as Vps21 small GTPase are delivered to the membrane of newly formed signaling endosomes, completing the signaling endosome biogenesis. Error bars represent mean ± SEM. More than 100 cells were analyzed for three biological replicates. Data points are color-coded according to biological replicates. Scale bars: 5 μm. The numbers next to blots denote molecular weights (kDa). Source data are available for this figure: SourceData F6.

vacuole-retaining motif. Another possibility is that distinct microdomains of the vacuolar membrane encode the fates of resident proteins. During prolonged nutrient starvation, two distinct microdomains of the vacuolar membrane become visible, one enriched with and the other depleted of sterols (Toulmay and Prinz, 2013; Kim and Budin, 2024). Interestingly, the sterol-enriched raft-like microdomain accommodates proteins including the EGO complex and Ivy1, similar to signaling endosomes. On the other hand, the v-ATPase resides in the sterol-poor microdomain. If in unstarved cells these microdomains also exist or are temporarily/locally formed during signaling endosome biogenesis, then signaling endosomes might originate from the sterol-rich domain, resulting in the selective inheritance of vacuolar proteins residing there. Interestingly, our Airyscan microscopy visualized distinct, uneven decoration of the vacuolar membrane by Tor1 and Vph1 (Fig. 3 F enlarged panels). Whether these patterns represent membrane microdomains could be investigated in the future.

An equally important question is how signaling endosomes acquire endosomal proteins such as the Vps21 GTPase (Fig. 3 D). These proteins could be individually delivered to signaling endosomes or arrive by membrane fusion. The latter scenario would be consistent with the recently proposed model that a fraction of canonical endosomes mature into signaling endosomes (Füllbrunn et al., 2024) if maturation involves fusion with vacuole-derived nascent signaling endosomal membrane. Further support for the fusion scenario is the localization of Dip5, a transmembrane cargo of the endocytic pathway, to signaling endosomes (Fig. S5 D). In this context, the Ivy1-positive, Tor1-negative subpopulation of endosomes may serve as a precursor of signaling endosomes together with the vacuole-derived membrane (Fig. S2 G). Collectively, this and previous studies position signaling endosomes at the intersection of the "forward" endocytic flow from the plasma membrane and the "reverse" flow that initiates from the vacuoles. The merger of the two flows would explain the vacuole–endosome hybrid protein/lipid composition of signaling endosomes.

The physiological roles of multiple nutrient transporters residing at signaling endosomes (Fig. 5 D) remain to be investigated. It is possible that signaling endosomes work as nutrient storage as vacuoles do, though this scenario is counterintuitive given the low abundance and small size of signaling endosomes. Alternatively, these transporters may function as nutrient receptors ("transceptors") and signal to TORC1 as the mammalian SLC38A9 amino acid transceptor does (Rebsamen et al., 2015; Wang et al., 2015). Yet another possibility is that these transporters are in the process of being trafficked from vacuoles to another intracellular location.

The roles of signaling endosomes in intracellular protein trafficking remain poorly understood. Atg18 mediates $PI(3,5)P_2$-dependent retrograde trafficking from the vacuolar membrane to canonical endosomes (Dove et al., 2004). Mechanistic details of this process are unknown, in particular, whether this trafficking initiates from a specific site of the vacuolar membrane. The role of Atg18 in signaling endosome formation discovered here raises the possibility that signaling endosomes provide the route for retrograde trafficking. If so, the selective migration of vacuolar proteins to newly formed signaling endosomes could serve as a cargo selection process (Fig. S2 G). A fusion event between the vacuole-derived membrane and canonical endosomes might be central for Atg18-mediated retrograde trafficking.

Another mystery is the reason for the heterogeneity within a cell population. It is unclear why only a fraction of cells form signaling endosomes even when Atg18 expression is synchronized (Fig. 3 B). The correlation with the budding status provides a partial answer (Fig. S4, C and D), warranting further study to elucidate exactly in which phases cell cycle progression influences the signaling endosome formation and how. As there is still heterogeneity among cells with the same budding status, there must exist additional unknown factors impacting signaling endosome formation. Possible cues include cellular replication age and fluctuation of metabolic states (and thus TORC1 activity) at the single-cell level. Notably, TORC1, Fab1, and Atg18 form a positive feedback loop (Fig. 6 D). Therefore, a small, possibly spontaneous increase in the concentration or activity of these molecules may nonlinearly accelerate the biogenesis of signaling endosomes once passing a certain threshold.

The positive feedback loop involving TORC1, Fab1, and Atg18 may explain why cells typically form only a single signaling endosome at a time (Fig. S2 A). The accelerated signal amplification through this feedback loop may allow a single site for signaling endosome formation to survive through competitive blocking of other precursor sites on the vacuolar membrane. This putative mechanism would be analogous to how the singularity of cellular polarization is ensured by the positive feedback regulation of the Cdc42 small GTPase (Howell et al., 2009; Freisinger et al., 2013). Mathematical modeling approaches could help examine this scenario.

The present study provides our first insight into signaling endosome biogenesis, opening many fundamental questions. Previous studies have identified several protein machinery controlling the abundance and/or the signaling function of signaling endosomes, including the Fab1, HOPS, and ESCRT complexes (Chen et al., 2021; Gao et al., 2022). A limitation of observing steady-state cell populations however meant that these studies did not discriminate effects on the initial formation, maintenance, and degradation of signaling endosomes. The Atg18 synchronization system we have developed here allows us to

dissect each step over time, making it possible to investigate the dynamic life cycle of signaling endosomes.

## Materials and methods

### Yeast strains, plasmids, and growth conditions

*Saccharomyces cerevisiae* yeast strains and plasmids used in this study are listed in Tables S1 and S2, respectively. Yeast strains were made prototrophic with indicated plasmids and/or empty vector plasmids and grown in synthetic complete media without uracil, leucine, and histidine (SC-ULH; SD Broth/2% Glucose [CSM0210; FORMEDIUM] plus Complete Supplement Mixture Triple Drop-Out -His, -Leu, -Ura [FORMEDIUM DCS0991]).

As exceptions, in Fig. S3 E and Fig. S5, yeast strains were transformed with pRS415-*mCherry-GTR1* and grown in synthetic complete media without leucine (SC-L; SD Broth/2% Glucose [CSM0210; FORMEDIUM] plus Complete Supplement Mixture Triple Drop-Out -Leu [DCS0091; FORMEDIUM]).

For TORC1 activation assays, yeast strains were initially grown in a synthetic dextrose–proline medium (0.19% yeast nitrogen base without amino acids and without ammonium sulfate [CYN0501; FORMEDIUM], 0.05% proline, and 2% glucose) and stimulated with glutamine or aspartate for 2 min.

For the GFP-Atg8 cleavage assay, cells were grown to the mid-log phase in SC-ULH and treated with rapamycin for 2 h to induce autophagy.

For the *GAL* shut-off experiment, cells were first cultured in synthetic raffinose media and then in synthetic galactose media for 3 h. After being washed, cells were cultured in SC-ULH media for 3 h before imaging.

In all experiments, yeast strains were grown to the mid-log phase at 30°C.

### Fluorescence microscopy

Images of live fluorescent cells were captured with a VoX spinning disk confocal microscope (Perkin Elmer) equipped with a Flash 4.0V3 camera and a PlanApo VC 60×/1.4 Oil DIC N2 objective.

For time-lapse imaging (Fig. 3, E and F; Fig. 4 A; Fig. 5 B; and Fig. S3 G), live cells were immobilized on a glass-bottom dish (81158; ibidi) coated with 1 mg/ml concanavalin A (C2010; Sigma-Aldrich). After being washed at time (t) = 0, cells were treated with SC-ULH media containing the indicated final concentration of aTc (10009542; Cayman Chemicals). Images were obtained with a ZEISS LSM880 confocal microscope (Carl Zeiss) equipped with a PlanApo 63×/1.4 Oil DIC M27 objective and ZEN black software (Carl Zeiss) and z-projected. The temperature was kept at 30°C throughout the time course. FM4-64 (T13320; Invitrogen) staining was performed as previously described (Hatakeyama et al., 2019) prior to cell immobilization.

Images were processed using Fiji-ImageJ software (National Institutes of Health, https://imagej.net/software/fiji/). For the budding status analysis, after individual cells were segmented by Cellpose (Stringer et al., 2021), cell diameters were measured by Fiji. The ratio between the diameter of daughter cells and that of mother cells was calculated. Cells were classified as large- or small-budded with the threshold at 0.5.

### Western blotting

Yeast cells were treated with 6.7 % trichloroacetic acid (final concentration), pelleted, washed with 70 % ethanol, dissolved in urea buffer (50 mM Tris-HCL [pH 8.0], 5 mM EDTA, 6 M urea, 1% SDS, Pefabloc, and PhosSTOP), and disrupted with glass beads using a FastPrep-24 homogenizer (MP Biomedical). Samples were heated at 65°C for 10 min in Laemmli SDS sample buffer and subjected to SDS-PAGE and immunoblotting experiments. The following antibodies were used: mouse anti-FLAG (1: 5,000, F1804; Sigma-Aldrich), rabbit anti-Tor1 (1:1,000, gifted from T. Maeda), rabbit anti-Sch9-pThr$^{737}$ (1:10,000, gifted from C. De Virgilio), goat anti-Sch9 (1:1,000, gifted from C. De Virgilio), mouse anti-GFP (1:1,000, 11814460001; Roche), goat anti-mouse IgG-HRP (1:3,000, Invitrogen 626520), goat anti-rabbit IgG-HRP (1:3,000, A16096; Invitrogen), and donkey anti-goat IgG-HRP (1:3,000, PA128664; Invitrogen).

### Statistical analysis

Two-sided Welch's *t* tests were used to calculate P values (unpaired tests for data points in black; paired tests for color-coded data points) except for Fig. S4, C and D. The P values with an asterisk were less than the significance levels (alpha = 0.05) corrected by the Holm–Sidak method. For Fig. S4, C and D, Tukey's multiple comparisons were used.

### Online supplemental material

Fig. S1 shows the functionality of tagged protein constructs. Fig. S2 shows an extended analysis of signaling endosomes. Fig. S3 shows further evidence of CROP-mediated signaling endosome biogenesis. Fig. S4 shows additional data from Atg18 induction experiments. Fig. S5 shows the search for additional signaling endosomal proteins. Table S1 shows the yeast strains used in this study. Table S2 shows the plasmids used in this study.

### Data availability

The data is available from the corresponding author upon reasonable request.

## Acknowledgments

We thank Claudio De Virgilio (University of Fribourg, Fribourg, Switzerland) and Christian Ungermann (University of Osnabrück, Osnabrück, Germany) for yeast strains, plasmids, and for helpful suggestions; Takahiro Shintani (Tohoku University, Sendai, Japan) for plasmids and helpful suggestions; Maya Schuldiner (Weizmann Institute of Science, Rehovot, Israel) and Michael Knop (Heidelberg University, Heidelberg, Germany) for yeast strain libraries; Takeshi Noda (Osaka University, Suita, Japan), Satoshi Okada (Kyushu University, Fukuoka, Japan, National BioResource Project plasmid BYP9806), and Joerg Stelling (ETH Zürich, Zürich, Switzerland, plasmid FRP2350/ 2365; Addgene) for plasmids; Tatsuya Maeda (Hamamatsu University School of Medicine, Hamamatsu, Japan) for the anti-Tor1 antibody; Daniel Paterson, Patryk Marcinkowski, Eri Hirata, and Saran Babooraj (previously University of Aberdeen) for their contribution to the visual screening using YeastRGB; members of the Hatakeyama lab and Chromosome & Cellular

Dynamics Section (University of Aberdeen) and Tokai TOR Conference for discussion; Megan Robertson, Colin Ferguson, Arrosan Rajalingam, and Microscopy and Histology Core Facility members (University of Aberdeen) for technical support; Anne Donaldson (University of Aberdeen) for advising on the manuscript.

This work was funded by the Biotechnology and Biological Sciences Research Council (BB/V016334/1 and BB/X018229/1 to R. Hatakeyama) and the University of Aberdeen. K. Muneshige is a recipient of the Japan Student Services Organization scholarship. Open Access funding was provided by the University of Aberdeen.

Author contributions: K. Muneshige: Conceptualization, Data curation, Formal analysis, Investigation, Methodology, Validation, Writing - original draft, R. Hatakeyama: Conceptualization, Funding acquisition, Supervision, Visualization, Writing - original draft, Writing - review & editing.

Disclosures: The authors declare no competing interests exist.

Submitted: 2 July 2024

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

# Supplemental material

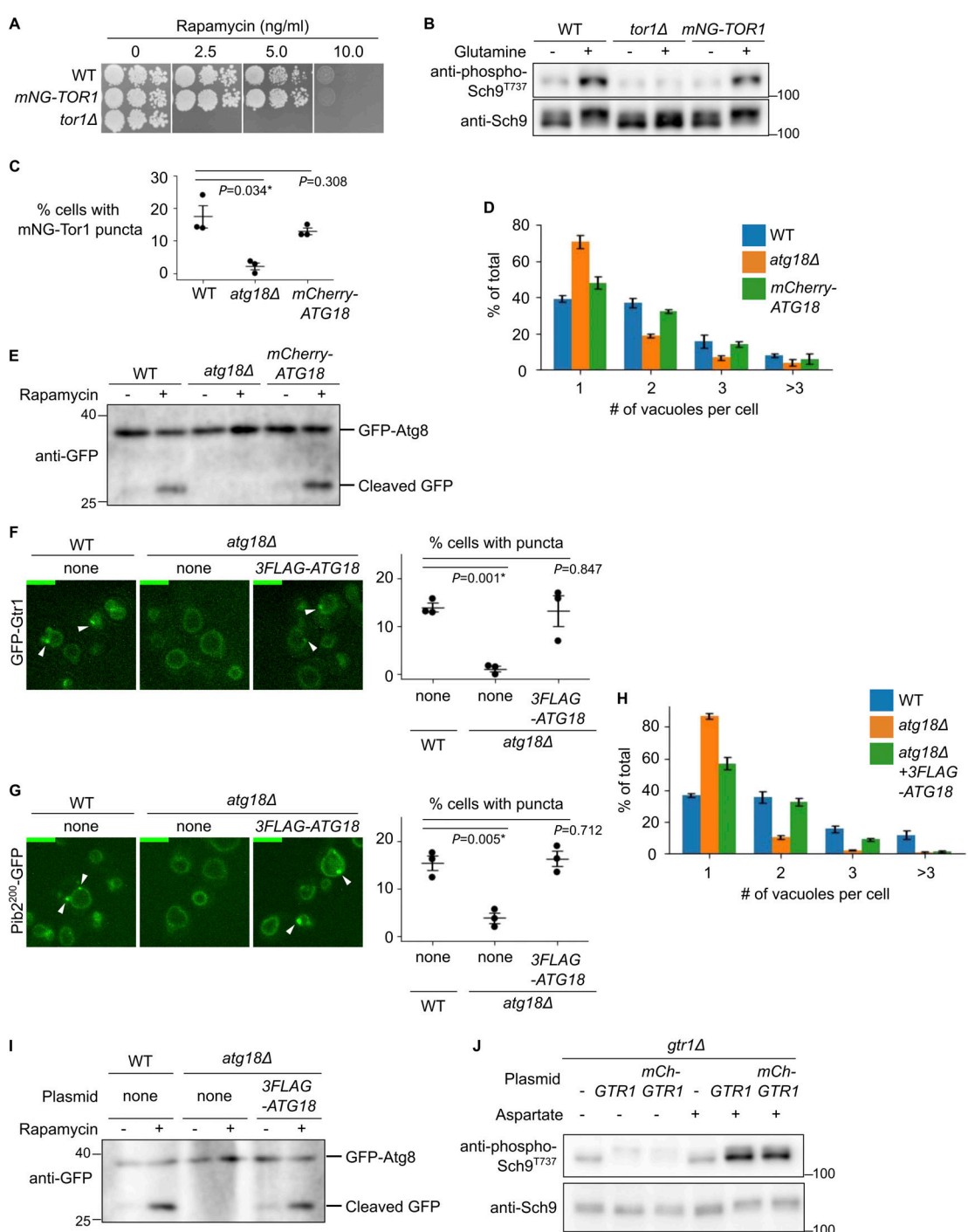

Figure S1. **Functionality of tagged protein constructs. (A and B)** Functionality of mNeonGreen-Tor1 (mNG-Tor1) used in this study. **(A)** Serial dilutions of the indicated strains were spotted and grown on YPD agar plates containing rapamycin at the indicated final concentration. **(B)** The indicated strains were stimulated or not for 2 min with 3 mM glutamine. The levels of Thr[737]-phosphorylated Sch9 and the total Sch9 were analyzed by western blotting. **(C–E)** Functionality of mCherry-Atg18 used in this study. **(C and D)** The indicated strains expressing mNG-Tor1 were analyzed by fluorescent microscopy. Quantifications of puncta-containing cells (C) and the number of vacuoles per cell (D) are shown. **(E)** The indicated strains expressing GFP-Atg8 were treated, or not, with 200 ng/ml rapamycin and analyzed by western blotting. **(F–I)** Functionality of 3FLAG-Atg18 used in this study. **(F)** The indicated strains expressing GFP-Gtr1 were analyzed by fluorescence microscopy. Arrowheads indicate signaling endosomes. Quantifications of puncta-containing cells are shown on the right. **(G)** The indicated strains expressing Pib2[200]-GFP were analyzed by fluorescent microscopy. Arrowheads indicate signaling endosomes. Quantification of puncta-containing cells is shown on the right. **(H)** The indicated strains expressing Pib2[200]-GFP were analyzed by fluorescent microscopy. The quantifications of the number of vacuoles per cell are shown. **(I)** The indicated strains expressing GFP-Atg8 were treated, or not, with 200 ng/ml rapamycin and analyzed by western blotting. **(J)** Functionality of mCherry-Gtr1 used in this study. gtr1Δ cells expressing the indicated plasmids were stimulated or not for 2 min with 0.1 mM aspartate. The levels of Thr[737]-phosphorylated Sch9 and the total Sch9 were analyzed by western blotting. Error bars represent mean ± SEM, >100 cells were analyzed for three independent biological replicates. Scale bars: 5 µm. The numbers next to blots denote molecular weights (kDa). Source data are available for this figure: SourceData FS1.

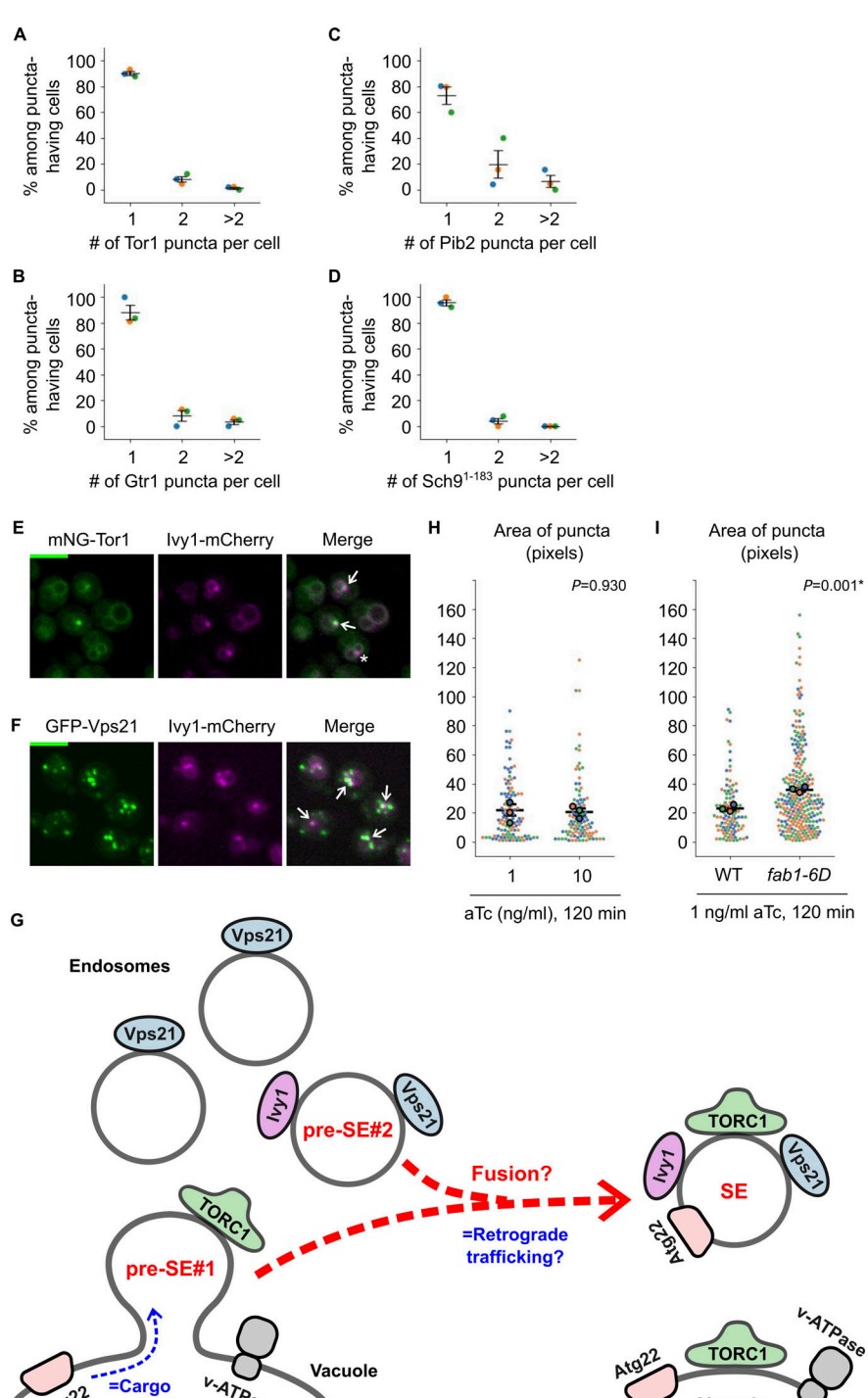

Figure S2. **Extended analysis of signaling endosomes. (A–D)** Distribution of the number of signaling endosomes per cell. Wild-type cells expressing the indicated fusion proteins were analyzed by fluorescent microscopy. Data points are color-coded according to biological replicates. **(E)** Colocalization of Tor1 and Ivy1. Cells co-expressing mNeonGreen-Tor1 (mNG-Tor1) and Ivy1-mCherry were analyzed by fluorescent microscopy. Arrows indicate colocalization of Tor1 and Ivy1 at signaling endosomes. The asterisk indicates an Ivy1-containing structure devoid of Tor1. **(F)** Colocalization of Vps21 and Ivy1. Cells co-expressing GFP-Vps21 and Ivy1-mCherry were analyzed by fluorescent microscopy. Arrows indicate colocalization of Vps21 and Ivy1. **(G)** Hypothetical merger with the endocytic pathway at a later step of signaling endosome biogenesis. In this model, the vacuole-derived membrane (pre-SE#1) fuses with the Ivy1-positive pool of canonical endosomes (pre-SE#2) to complete signaling endosome biogenesis. See the main text for details. SE: signaling endosome. pre-SE: precursor of signaling endosome. **(H)** No difference in the size of Tor1 puncta induced by different concentrations of aTc. $WTC_{846}$-ATG18 cells expressing mNG-Tor1 were treated with aTc at the indicated final concentrations and analyzed by fluorescent microscopy. **(I)** Bigger size of induced Tor1 puncta in *fab1-6D* strains. The indicated $WTC_{846}$-ATG18 cells expressing mNG-Tor1 were treated with 1 ng/ml aTc and analyzed by fluorescent microscopy. In H and I, quantifications of the apparent size (area) of Tor1 puncta (pixels) are shown. Small dots represent individual data points. Larger dots outlined in black are the averages for each biological replicate. Both the individual data points and averages are color-coded to distinguish biological replicates. Error bars represent mean ± SEM. More than 20 Tor1 puncta were analyzed for each biological replicate. Scale bars: 5 μm.

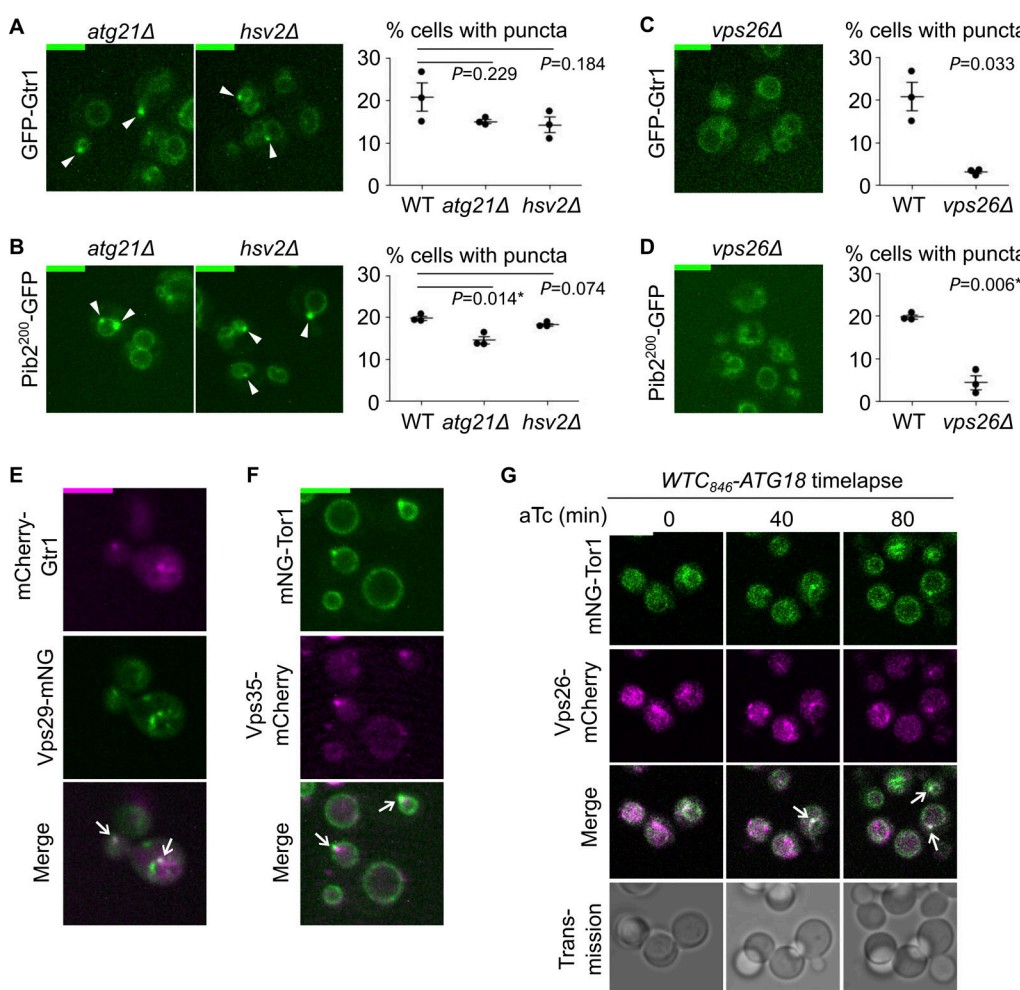

Figure S3. **Further evidence of CROP-mediated signaling endosome biogenesis. (A and B)** No requirement for other PROPPIN proteins for signaling endosome formation. The indicated strains expressing GFP-Gtr1 (A) or Pib2$^{200}$-GFP (B) were analyzed by fluorescent microscopy. Arrowheads indicate signaling endosomes. Quantifications of puncta-containing cells are shown on the right. **(C and D)** Requirement of Vps26 for signaling endosome formation. The indicated strains expressing GFP-Gtr1 (C) or Pib2$^{200}$-GFP (D) were analyzed by fluorescent microscopy. Quantifications of puncta-containing cells are shown on the right. **(E)** Localization of Vps29 to signaling endosomes. Cells co-expressing Vps29-mNeonGreen (Vps29-mNG) and mCherry-Gtr1 were analyzed by fluorescent microscopy. Arrows indicate colocalization of Vps29 and Gtr1 at signaling endosomes. **(F)** Localization of Vps35 to signaling endosomes. Cells co-expressing mNG-Tor1 and Vps35-mCherry were analyzed by fluorescent microscopy. Arrows indicate colocalization of Tor1 and Vps35 at signaling endosomes. **(G)** Localization of Vps26 to newly formed signaling endosomes. *WTC$_{846}$-ATG18* cells co-expressing mNG-Tor1 and Vps26-mCherry were treated with 1 ng/ml aTc and analyzed by time-lapse fluorescent microscopy. Arrows indicate colocalization of Tor1 and Vps26 at newly formed signaling endosomes. Error bars represent mean ± SEM. More than 100 cells were analyzed for each three independent biological replicates. Scale bars: 5 μm.

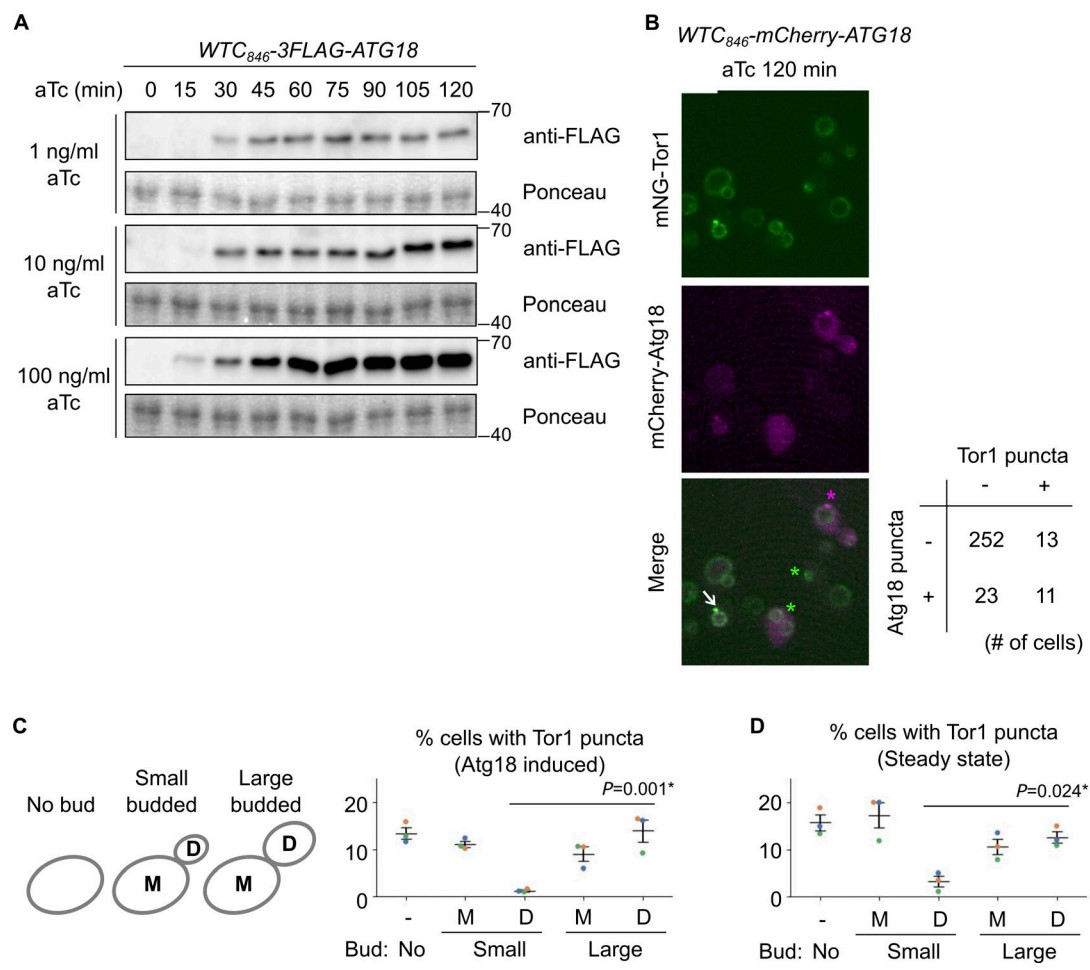

Figure S4. **Additional data from Atg18 induction experiments. (A)** Confirmation of WTC$_{846}$-induced Atg18 expression with different aTc concentrations at different time points. The *WTC$_{846}$-3FLAG-ATG18* cells were treated with the indicated final concentrations of aTc, collected every 15 min, and analyzed by western blotting. **(B)** The heterogeneity of Atg18 puncta formation and signaling endosome biogenesis in a cell population. *WTC$_{846}$-mCherry-ATG18* cells expressing mNeonGreen-Tor1 (mNG-Tor1) were treated with 10 ng/ml aTc for 120 min and analyzed by fluorescent microscopy. The arrow indicates the colocalization of Tor1 and Atg18 at newly formed signaling endosomes. Green asterisks indicate newly formed signaling endosomes devoid of the Atg18 signal. The magenta asterisk indicates an Atg18 puncta devoid of the Tor1 signal. The quantification of cells showing the (co)presence of Tor1 puncta and Atg18 puncta is shown in the table. **(C and D)** Correlation between signaling endosome formation and cell cycle progression. **(C)** Left: The scheme of cell categorization based on the budding status. M: mother cell, D: daughter cell. See also Materials and methods. Right: *WTC$_{846}$-ATG18* cells expressing mNG-Tor1 were treated with 1 ng/ml aTc for 120 min and analyzed by fluorescent microscopy. **(D)** Wild-type cells expressing mNG-Tor1 were analyzed by fluorescent microscopy. In C and D, cells were classified into the indicated five categories. Quantification of puncta-containing cells is shown. Error bars represent mean ± SEM. More than 50 Tor1 puncta were analyzed for each biological replicate. Data points are color-coded according to biological replicates. Scale bar: 5 µm. The numbers next to blots denote molecular weights (kDa). Source data are available for this figure: SourceData FS4.

## A  Amino acid transporters

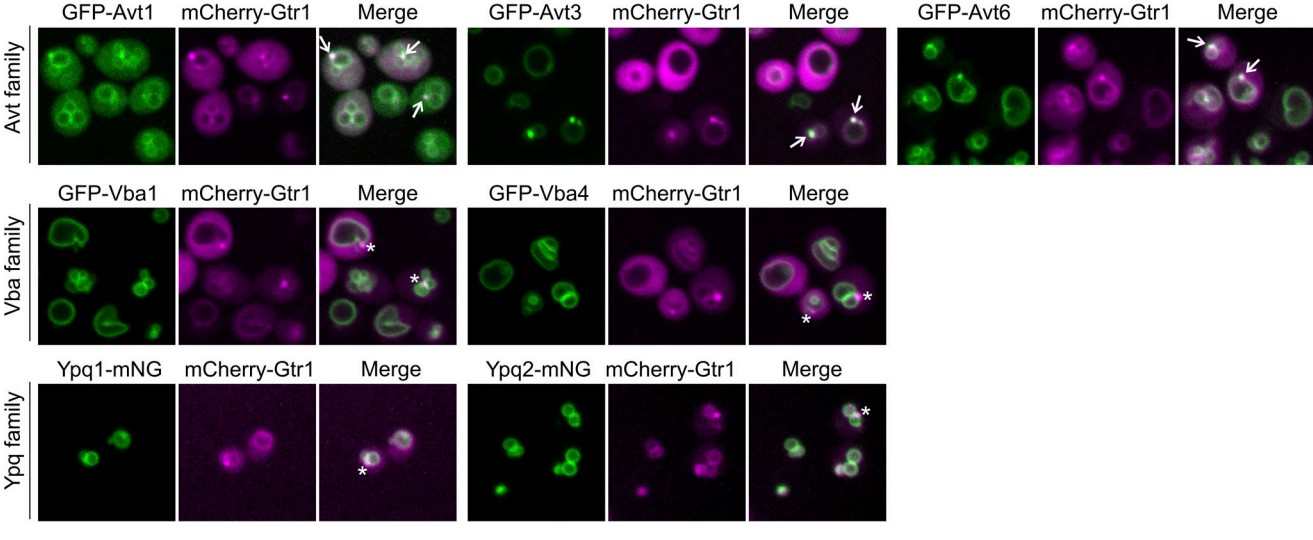

## B  AP-3 cargoes

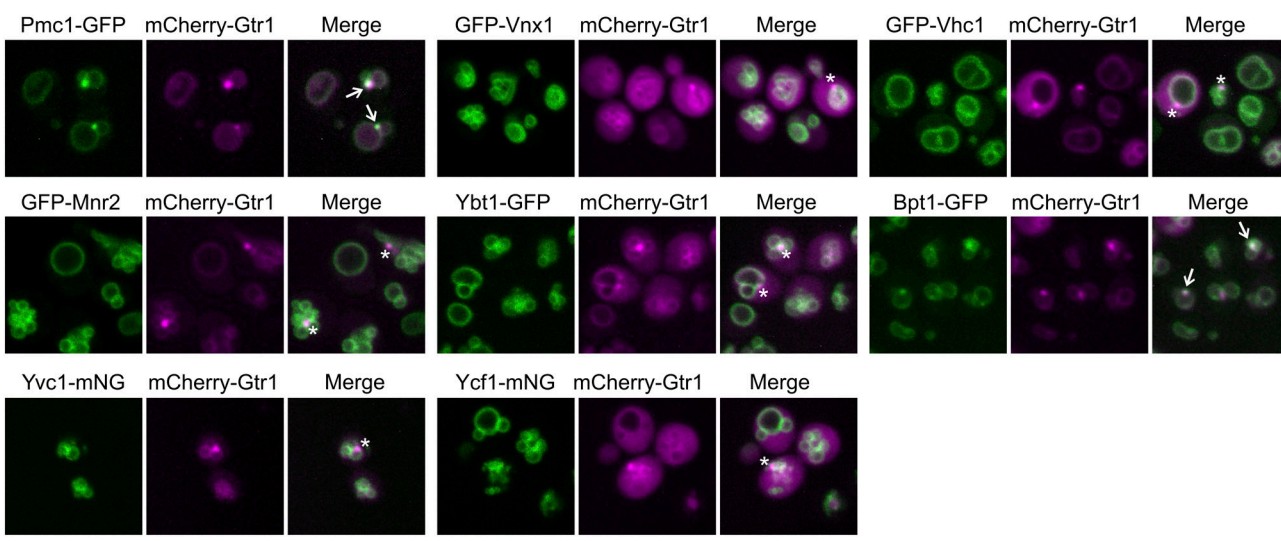

## C  Recycling cargo

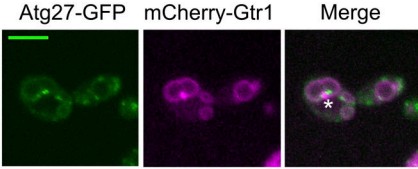

## D  Plasma membrane amino acid transporter

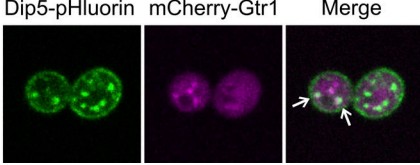

Figure S5.  **Search for additional signaling endosomal proteins. (A)** Localization of the AVT family amino acid transporters to signaling endosomes. Strains co-expressing the indicated GFP- or mNeonGreen (mNG)-fusion vacuolar amino acid transporters and mCherry-Gtr1 were analyzed by fluorescent microscopy. **(B)** Localization of some, but not all, AP-3 cargoes to signaling endosomes. Strains co-expressing the indicated GFP-or mNG-fusion AP-3 cargoes and mCherry-Gtr1 were analyzed by fluorescent microscopy. **(C)** Absence of Atg27 at signaling endosomes. Cells co-expressing Atg27-GFP and mCherry-Gtr1 were analyzed by fluorescent microscopy. **(D)** Localization of Dip5 to signaling endosomes. Cells co-expressing Dip5-pHluorin and mCherry-Gtr1 were analyzed by fluorescent microscopy. Arrows indicate colocalization of the examined proteins with mCherry-Gtr1 at signaling endosomes. Asterisks indicate the absence of the examined proteins on signaling endosomes. Scale bars: 5 μm.

Provided online are Table S1 and Table S2. Table S1 shows yeast strains used in this study. Table S2 shows plasmids used in this study.

