## [Peer Review File · The Journal of Cell Biology]

Vacuoles provide the source membrane for TORC1-containing signaling endosomes

Kenji Muneshige and Riko Hatakeyama

Corresponding Author(s): Riko Hatakeyama, University of Aberdeen

Review Timeline:

Submission Date:	2024-07-02
Editorial Decision:	2024-08-23
Revision Received:	2024-12-12
Editorial Decision:	2025-02-03
Revision Received:	2025-02-12

Monitoring Editor: Lois Weisman

Scientific Editor: Andrea Marat

Transaction Report:

DOI: <https://doi.org/10.1083/jcb.202407021>

August 23, 2024

Re: JCB manuscript #202407021

Dr. Riko Hatakeyama
University of Aberdeen
Institute of Medical Sciences
Foresterhill
Aberdeen AB25 2ZD
United Kingdom

Dear Dr. Hatakeyama,

Thank you for submitting your manuscript entitled "TORC1-containing signaling endosomes source membrane from vacuoles". Your manuscript has been assessed by expert reviewers, whose comments are appended below. Although the reviewers express potential interest in this work, significant concerns unfortunately preclude publication of the current version of the manuscript in JCB.

You will see that the reviewers are somewhat split in their assessment of the suitability of your study for JCB, however they share many experimental concerns. Overall, we find that to be suitable for JCB it is essential that you provide better documentation about the upstream signaling for signaling endosome production as well as more clarification as to whether the signaling endosomes are a distinct entity or are part of the vacuole. Additional information about what other proteins are part of this organelle is also essential. The reviewers have all provided constructive suggestions on how these aims could be achieved.

For further consideration at JCB further substantiation of the original claims made in the manuscript are critical. However, while we would welcome additional data regarding the biological significance of signaling endosomes as suggested by one of the reviewers, this is not essential.

Please let us know if you are able to address the major issues outlined above and wish to submit a revised manuscript to JCB. Note that a substantial amount of additional experimental data likely would be needed to satisfactorily address the concerns of the reviewers. The typical timeframe for revisions is three to four months, but could be extended if necessary. Please note that papers are generally considered through only one revision cycle, so any revised manuscript will likely be either accepted or rejected based on the feedback of all original reviewers or suitable replacements if necessary.

If you choose to revise and resubmit your manuscript, please also attend to the following editorial points. Please direct any editorial questions to the journal office.

GENERAL GUIDELINES:

Text limits: Character count is < 40,000, not including spaces. Count includes title page, abstract, introduction, results, discussion, and acknowledgments. Count does not include materials and methods, figure legends, references, tables, or supplemental legends.

Figures: Your manuscript may have up to 10 main text figures. To avoid delays in production, figures must be prepared according to the policies outlined in our Instructions to Authors, under Data Presentation, <https://jcb.rupress.org/site/misc/ifora.xhtml>. All figures in accepted manuscripts will be screened prior to publication.

Supplemental information: There are strict limits on the allowable amount of supplemental data. Your manuscript may have up to 5 supplemental figures. Up to 10 supplemental videos or flash animations are allowed. A summary of all supplemental material should appear at the end of the Materials and methods section.

Please note that JCB now requires authors to submit Source Data used to generate figures containing gels and Western blots with all revised manuscripts. This Source Data consists of fully uncropped and unprocessed images for each gel/blot displayed in the main and supplemental figures. File names for Source Data figures should be alphanumeric without any spaces or special characters (i.e., SourceDataF#, where F# refers to the associated main figure number or SourceDataFS# for those associated with Supplementary figures). The lanes of the gels/blots should be labeled as they are in the associated figure, the place where cropping was applied should be marked (with a box), and molecular weight/size standards should be labeled wherever possible. Source Data files will be made available to reviewers during evaluation of revised manuscripts and, if your paper is eventually

published in JCB, the files will be directly linked to specific figures in the published article.

If you choose to resubmit, please include a cover letter addressing the reviewers' comments point by point. Please also highlight all changes in the text of the manuscript.

Regardless of how you choose to proceed, we hope that the comments below will prove constructive as your work progresses. We would be happy to discuss them further once you've had a chance to consider the points raised. You can contact the journal office with any questions at cellbio@rockefeller.edu.

Thank you for thinking of JCB as an appropriate place to publish your work.

Sincerely,

Lois Weisman, PhD
Monitoring Editor

Andrea L. Marat, PhD
Deputy Editor

Journal of Cell Biology

Reviewer #1 (Comments to the Authors (Required)):

In this manuscript, K. Muneshige and R. Hatakeyama developed an ATG18 gene expression system to induce the synchronized formation of signaling endosomes (SEs). They used this system to investigate the source of SE membranes as well as some of the molecular machinery involved. The authors first demonstrated that ATG18 is necessary for SE formation and subsequently employed this gene in their inducible SE formation system. They determined that ATG18 functions as part of the CROP complex to generate SEs, with the SEs inheriting their membrane from the vacuole. Additionally, the authors used the ATG18-inducible system to show that TORC1 and Fab1 signaling promote SE formation.

However, a key concern with this study is the function and biological significance of signaling endosomes, given that they exist in only ~15% of yeast cells. What signals generated by SEs are crucial for yeast? Under what conditions is the biogenesis of signaling endosomes triggered? And what are the consequences of losing signaling endosomes? Without addressing these questions, it is uncertain whether this study will resonate with the broader readership of the Journal of Cell Biology. Below are more specific comments.

Major Notes:

Fig 1g: Ivy1 was used in previous studies from the Ungermann group as a signaling endosome marker. In this paper, the authors report that their mNeonGreen-tagged Ivy1 localizes to other subcellular loci as well. Since this is a significant point of controversy, it requires further discussion/investigation. Is the difference due to the mNG tag? What are the identities of Ivy1-positive puncta if most of them are not signaling endosomes? A set of colocalization experiments with other endosomal markers should address this question. Additionally, a discussion on what confusion might have been generated by using Ivy1 as a signaling endosome marker is necessary.

Fig 2g: As an additional line of evidence supporting the retromer's involvement in signaling endosome formation, can the retromer components be co-localized with the signaling endosomes? Is it possible to show the recruitment of retromer components using time-lapse imaging after the induction of signaling endosome formation?

Fig 3c: The time-lapse imaging occurs over a period of 2 hours; however, no cell growth or division is observed. Normally, the doubling time of yeast cells is 1.5-2 hours, depending on the media. The lack of cell growth or division suggests that the yeast cells were severely stressed and/or that the imaging conditions were problematic. Could the signaling endosomes have been triggered by the stressful imaging conditions and therefore should be regarded an artefact of stress response?

Fig 4a: Similar to Fig 3c, little to no cell growth and division is observed over a 2-hour period. Additional controls are needed. In addition to mNG-Tor1, Vph1 could be used to show that SE formation selectively recruits mTOR while excluding Vph1, thus clearly demonstrating that the puncta formation is not due to stress-induced aggregation.

Lastly, this paper provides evidence that signaling endosomes are formed from the vacuole rather than at a branching point in the endocytic pathway. Since this model differs from previous models, the authors should emphasize and discuss these differences to provide greater clarity to the field. Previous studies reported the requirement of HOPS and ESCRT for SEs. How does this fit into the current model?

Minor Notes:

Fig 1b: The authors should provide one or two sentences explaining why a rapamycin survival assay was used to determine the

functionality of mNG-Tor1.

Fig 3b: The SEs from cells exposed to 10 ng/ml aTC are significantly larger compared to those exposed to 1 ng/ml. Phosphomimetic fab1-6D (Fig 5c) also seems to have larger SEs. What could be the reason for the size differences observed? Additional quantification and distribution analysis on SEs should be performed. In addition to assessing how many cells form puncta, how many puncta per cell are there? Are certain cells more likely to form SEs compared to others? Is SE formation a one-time event, or does it occur multiple times within the same cell?

Reviewer #2 (Comments to the Authors (Required)):

Kenji Muneshige and Riko Hatakeyama provide a nice study to show that Atg18, in concert with retromer, is required for the formation of signaling endosomes (SE). SEs are structures containing Torc1 and its regulators and the full function of these compartments and how they are generated are important to resolve. These investigators made a system to induce Atg18 and watch the formation of SEs. The process is slow (~60-70 min) but helpful enough to show that FM4-64 that had been previously chased to the vacuole can be found in the newly formed SEs. The authors are careful not to say that all SE membranes are derived from the vacuole, which is an important clarification. Though an explicit discussion of this point (that 'forward' traffic in the endosomal system may intersect in this compartment since these experiments do not rule that out) would be helpful for the reader. In general, the studies are nicely done and the main point substantiated. What distinguishes the SE as a unique endosome vs an outcropping of the vacuole with some Vps21/Rab5 on it is a matter of language perhaps, but gets at the heart of how to interpret the novelty of the central finding of this paper. As is, I think the work is valuable and may be instructive for JCB readers. Here is a list of improvements that should be considered.

- The time course of Atg18 induction is likely slow based on the previously published description of this tet-controlled promoter construct. This study needs to have a time course of Atg18 induction at the various levels of tet and not just a single time point of various levels of tet.
- A point the authors make is why SEs are only in a subset of cells. This is an intriguing point and observed in these Atg18-induction experiments and in normal cultures of WT cells. For this particular experiment, however, the finding would be greatly strengthened from showing induction of the functional mCherry-Atg18 protein with simultaneous imaging of a couple of the SE GFP markers to view throughout the population of cells to determine heterogeneity,
- Figure 4 shows the turn-on of Atg18. There is no quantitation of the number of compartments formed and the arrows actually look a bit arbitrary. For instance, you could draw arrows at different structures at the 0-30 time points that are probably bunched vacuolar membrane giving the illusion of an SE. This kinetic analysis needs multiple samples and a set criteria to define when there is an SE vs a blobby vacuole.
- Figure 4 needs better microscopy. Deconvoluted Z-stacks (or even maximum point projections) would be helpful - this would show dots through out the cell. The Zeiss LSM 880 used for these studies may even have Airyscan to help resolution.
- Atg18 production is probably fairly slow here, preventing more acute experiments such as finding whether retromer substrate membrane proteins pass through this compartment on their way back to the TGN. Yet, even with slow Atg18 production, there is a missed opportunity to examine the order of recruitment to the forming SE. Things like Pib2, PI3,5, Torc1, Vps21, etc, may arrive in a different order. Provided there is a quantitative assay that considers several samples that can smooth the estimate of arrival times, this would really strengthen this type of biogenesis story. This could be informative and surprising - for instance, in the one set of micrographs provided, Atg22 seemingly arrives in concentrated areas before Tor1 (Fig 4).
- There is a lot of speculation about what might be special about Atg22 and its distribution in SE. This looks like the only protein besides Vph1 that was looked at, so speculation in this regard warrants inspection of a larger set of proteins - especially those that one might expect to recycle from vacuolar membranes at some point.
- The authors state: "This membrane source for signaling endosomes differs starkly from our current knowledge of canonical endosomes, for which the membrane is supplied mainly from the plasma membrane and the Golgi apparatus via intracellular vesicle trafficking." There are no data in this manuscript showing that this does not happen for SEs - there is no test of 'forward' trafficking. Moreover, the idea that there is protein and lipid recycling from late endosomes and lysosomes back to early endosomal/TGN compartments - even routes dependent on Atg18 - have been articulated before so this claim is a bit exaggerated.
- The authors state: "This unique feature suggests that signaling endosomes should be most correctly viewed as a distinct organelle rather than a mere subpopulation of endosomes." This is helpful. Yet one can also consider the SE as a subdomain of the vacuolar compartment, leaving its communication with FM4-64 labelled vacuoles more to be expected. There is also the possibility (just from the ATG22 result) that SEs serve as a signaling center and also as a sorting station, similar to ER exit sites where lipids and proteins such as Atg22 accumulate for carrier packaging.

- Fix the reference formatting (relevant ones within a single set of '()').

Reviewer #3 (Comments to the Authors (Required)):

The manuscript by Muneshige and Hatakeyama represents an exciting advance in our understanding of signaling endosomes and TORC1 regulation. Many of the experimental approaches used in this paper are compelling, including the development and implementation of a new system to monitor signaling endosome biogenesis and the use of the tunable expression system for Atg18. The manuscript is clear and concise in its delivery, making the paper a joy to read. The novel finding that signaling endosomes arise from vacuoles in a manner dependent upon Atg18 and retromer is not only thought provoking but to my knowledge represents a new paradigm for regulation of membrane proteins at the vacuole. I am highly enthusiastic about this manuscript and overall, the quality of the data are very good. However, I have a few points for the authors to consider that may help bolster their claims.

Major Points:

1. Atg18 is required for signaling endosome formation but does not disrupt other endosomal compartments: This is the main takeaway from Figure 1, which is nicely presented. However, given the interplay between Golgi and endosomal compartments, it might be wise to also examine the distribution of a Golgi marked compartment (i.e., Sec7) to ensure it is unaltered in the *atg18Δ* cells. In addition, in Figure 1J, it would be useful to define the % of puncta co-localizing between mNG-Tor1 and mCherry-Atg18 (Pearsons and Manders coefficients). Finally, in Figure 1J the vacuoles appear somewhat enlarged which raises the question on how the function of the mCherry-tagged Atg18 was assessed. It would be wonderful if the data demonstrating that mCherry-Atg18 is functional was included as a supplement.

2. Icy1 continues to form puncta in the absence of Atg18: Some additional comment on why this is the case is warranted in the text of the manuscript. Does this suggest that Icy1 might be a precursor to Atg18 recruitment during formation of signaling endosomes?

What is known about how Icy1 is recruited to vacuole membranes? The fact that the authors indicate that the mNG-Icy1 specifically may be localizing to subpopulations of endosomes raises the question of how the mNG-Icy1 localization compares to other tagged forms of Icy1. Please address these issues in the text; Past citations may suffice to clarify what is occurring.

3. Atg18 membrane scission function is required for formation of Tor1-containing signaling endosomes: In Figure 2A, only Tor1 puncta are monitored in the *hsv2* and *atg21* cells. It would be useful here to look at other signaling endosome markers to make the finding more compelling. In Figure 2F, the T56E and FG3G variants of 3FLAG-Atg18 are not expressed as well as the WT or Sloop mutants. The statement on line 166 of page 5 should be revised to reflect that difference; while these variants were expressed, they were expressed at a lower level than WT Atg18 (2-3 fold lower? Consider quantifying). These Atg18 mutants are reported to impact membrane scission, but the Atg18 lipid transfer activities, as part of Atg2-9-18 complex, should be intact. Perhaps assessing this facet of Atg18 function would be beneficial here (or citing where this has been shown previously). Similar to my comment for the mCherry-Atg18, what assays - other than the formation of the mNG-Tor1 puncta -- were done to ensure that the FLAG tagged Atg18 employed was functional? I suggest that monitoring vacuole morphology and localization/abundance of Gtr1 and/or Pib2 puncta would be helpful orthogonal approaches that could bolster Figure 2.

4. Atg18 is sufficient to allow for Tor1 puncta formation in add back experiments: These are exciting findings that are foundational to this study and so should be highly rigorous. In Figure 3B, the time course results for the % puncta formation has no error bars or statistical measures to ensure that the change from a few percent (perhaps increase the number of tick marks on your scale bar here please) to a maximum of 15% is statistically different. I realize that this requires considerable effort, but in my view it is required for the result to be compelling. The absence of Vph1 in the mNG-Tor1 puncta and the co-localization of mCherry-Vps21 with mNG-Tor1 puncta in Figures 3D and 3E should have some kind of measurement added (i.e., pearsons and manders).

5. Atg18 drives formation of a distinct, vacuole-derived, compartment: A critical aspect of the study will be demonstrating that the Atg18 puncta formed in WTC846 expression system are discrete from the vacuole membrane. To address this comment likely requires 3D imaging or EM. This is especially important as the claim is being made in the discussion that these are discrete organelles. The FM imaging nicely shows that the membrane protrusion from the vacuole co-localizes with Tor1, but it does not demonstrate that the Tor1 puncta is separate from the vacuole. From the data presented, it is hard to discern that the signaling endosomes marked by Tor1 are distinct compartments from the vacuole itself. Is there 3D imaging or an EM reference that can help distinguish between the possibilities that (i) the signaling endosome is a sub-compartment of the vacuole or (ii) the signaling endosome is a free and distinct membrane bound compartment?

Minor points

A) In Figure 2D, the loss of retromer most likely disrupts endosome formation in general. Is the loss of retromer here having a direct or indirect effect on signaling endosome formation?

B) In Figure 4, there appear to be other concentrated signal points for Tor1 that do not go on to form as robust an accumulation. This made me wonder if, like endocytic sites, there is a critical concentration of factors that needs to be assembled on the vacuole to generate these signaling endosomes. –Some mention of the smaller clusters of Tor1 signal might be useful.

C) What is the fate of the signaling endosome? Once formed from the vacuole, where does it go? Is it destined for fusion with the PM like other vesicles formed from lysosomes/vacuoles? Does it fuse to other compartments or does it return to the vacuole at some point? Some comment on this in the discussion seems warranted.

D) Text suggested changes -

Line 50 on page 3 - Signaling endosomes accommodate - consider revising to accumulate

Lines 53 and 56 on page 3 - no need to have 'its' in either of these sentences

Line 292-293 on page 8 - Biogenesis is repeated.

We thank the reviewers for their enthusiastic comments and constructive suggestions. We thoroughly addressed the points raised by reviewers, mostly by carrying out new experiments or otherwise by modifying texts. As a result, we believe that the quality of our manuscript has significantly improved.

In particular,

- Regarding what proteins are transferred from the vacuolar membrane to signaling endosomes, our manuscript now includes a list of 18 vacuolar proteins for which signaling endosomal localization has been either confirmed or denied (Figures 5D and S5A to S5C). This new dataset confirms the selective nature of protein transfer from vacuoles to signaling endosomes.
- We also show that the plasma membrane transporter Dip5 passes through signaling endosomes (Figure S5D). This data supports our argument (which was speculative in the original submission) that signaling endosomes form an intersection site for “retrograde” trafficking from the vacuoles with the “forward” flow of the endocytic pathway. We now visualize this concept in Fig S2G.
- Together, these new data have significantly expanded our knowledge of the protein composition (and thus potential biological functions) of signaling endosomes.

- We now show Airyscan high-resolution images of signaling endosomes (Figure 3F, see the enlarged panels). To our best knowledge, these are the finest live-cell images yet produced of signaling endosomes (and perhaps TORC1 localization in general).
- These Airyscan images visualize the signaling endosomes to be spherical structures located near the vacuolar membrane surface, arguing against the idea that signaling endosomes are a domain of the vacuolar membrane.
- We now show that the lipophilic dye FM4-64 does not readily migrate from vacuoles to mature signaling endosomes (Figure 4B), further establishing that signaling endosomes are a separate organelle.
- While these additional results point towards signaling endosomes being separate organelles, we recognize that we cannot completely exclude the possibility that an edge of signaling endosome maintains a connection to the vacuolar membrane. For clarification, we now explicitly discuss this possibility.

- As an unexpected outcome, Airyscan microscopy revealed distinct, uneven decoration of the vacuolar membrane by Tor1 and Vph1 (Figure 3F, see the enlarged panels), suggestive of distinct microdomains. This is a significant discovery even though it may not directly fall within the scope of this study. We mentioned this finding in the Discussion.

- As to why not all cells form signaling endosomes, we discovered a correlation between signaling endosome formation and cell cycle progression (Figure S4C and S4D). We also expanded our discussion on additional cues that may cause the population heterogeneity.

In addition, we performed the following experiments to solidify our conclusions.

- We confirmed the functionality of tagged proteins (by multiple assays; see Figure S1).
- We performed additional analyses of signaling endosomes including examining the distribution of their abundance per cell, their size distribution, and their relation to Ivy1-marked structures (Figure S2).
- We confirmed the role of the CROP complex in signaling endosome formation via orthogonal approaches (Figure S3).
- We performed additional experiments with our Atg18 induction system including the determination of time/dose-dependency of tetracycline stimulation, and a population analysis of induced Atg18 puncta formation plus its relation to signaling endosome formation (Figure S4).
- We generally improved the data quality by adding replicates, quantifications, and statistical tests.

Please find below the point-by-point responses to the reviewers' comments. Attached at the end of this document is the track-changed manuscript highlighting all the changes made from the initial submission.

Reviewer #1 (Comments to the Authors (Required)):

In this manuscript, K. Muneshige and R. Hatakeyama developed an ATG18 gene expression system to induce the synchronized formation of signaling endosomes (SEs). They used this system to investigate the source of SE membranes as well as some of the molecular machinery involved. The authors first demonstrated that ATG18 is necessary for SE formation and subsequently employed this gene in their inducible SE formation system. They determined that ATG18 functions as part of the CROP complex to generate SEs, with the SEs inheriting their membrane from the vacuole. Additionally, the authors used the ATG18-inducible system to show that TORC1 and Fab1 signaling promote SE formation.

We thank the reviewer for the detailed reading and constructive suggestions.

However, a key concern with this study is the function and biological significance of signaling endosomes, given that they exist in only ~15% of yeast cells. What signals generated by SEs are crucial for yeast? Under what conditions is the biogenesis of signaling endosomes triggered? And what are the consequences of losing signaling endosomes? Without addressing these questions, it is uncertain whether this study will resonate with the broader readership of the Journal of Cell Biology.

We addressed the biological significance of signaling endosomes in previous studies. We did so by (1) characterizing the *tor1^{D330}-3GFP* mutant strain that largely lacks signaling endosomes, (2) identifying specific substrates of the TORC1 pool residing at signaling endosomes, and (3) mutating phosphorylation sites on these substrates. As described in the Introduction section of this manuscript, lack of signaling endosomes or of local TORC1 signaling causes dysregulation of macro- and micro-autophagy and PI(3,5)P₂ synthesis. The present work was built upon these previous findings and specifically focuses on the biogenesis of signaling endosomes.

Furthermore, new experiments included in this revised manuscript version provide the following novel insights into the biology of signaling endosomes.

- We now found multiple nutrient transporter proteins residing at signaling endosomes (Figure 5D), hinting at the functions of signaling endosomes in nutrient transport and/or sensing.
- We now found that signaling endosome formation correlates with cellular budding status (Figure S4C and S4D), suggesting a potential link to cell cycle progression.

Below are more specific comments.

Major Notes:

Fig 1g: Ivy1 was used in previous studies from the Ungermann group as a signaling endosome marker. In this paper, the authors report that their mNeonGreen-tagged Ivy1 localizes to other subcellular loci as well. Since this is a significant point of controversy, it requires further discussion/investigation. Is the difference due to the mNG tag? What are the identities of Ivy1-positive puncta if most of them are not signaling endosomes? A

set of colocalization experiments with other endosomal markers should address this question.

We now show that lvy1-positive structures are indeed Vps21-positive endosomes (Figure S2F), which we suspect are precursors of signaling endosomes (Figure S2G).

This caveat of lvy1 as a marker for signaling endosomes is not limited to our mNeonGreen-tagged construct. We noticed that Christian Ungermann's group addressed the same issue in response to the peer-reviewers' comments on their 2022 JCB paper (PMID: 35404387). As detailed in the published Review History of this paper, they reached the same conclusion that lvy1 also localizes to other endosomal subpopulations.

Additionally, a discussion on what confusion might have been generated by using lvy1 as a signaling endosome marker is necessary.

As written above, the Ungermann group rigorously addressed this issue during their paper revision. They consolidated their lvy1-based findings by repeating them using Kog1 (a TORC1 subunit) as a genuine signaling endosome marker. Hence, we believe no major conclusions made in the past are compromised because of the lvy1 matter.

Even so, we agree it is beneficial to revisit and independently confirm the lvy1 caveat in this paper, as in Figs. S2F and S2G, because the misconception of seeing lvy1 as a specific marker for signaling endosomes seems to persist in the field.

Fig 2g: As an additional line of evidence supporting the retromer's involvement in signaling endosome formation, can the retromer components be co-localized with the signaling endosomes? Is it possible to show the recruitment of retromer components using time-lapse imaging after the induction of signaling endosome formation?

We have now done this, confirming the retromer colocalization with signaling endosomes in both snap-shot experiments (Figure 2G, S3E and S3F) and time-lapse imaging (Figure S3G).

Fig 3c: The time-lapse imaging occurs over a period of 2 hours; however, no cell growth or division is observed. Normally, the doubling time of yeast cells is 1.5-2 hours, depending on the media. The lack of cell growth or division suggests that the yeast cells were severely stressed and/or that the imaging conditions were problematic. Could the signaling endosomes have been triggered by the stressful imaging conditions and therefore should be regarded an artefact of stress response?

Fig 4a: Similar to Fig 3c, little to no cell growth and division is observed over a 2-hour period. Additional controls are needed. In addition to mNG-Tor1, Vph1 could be used to show that SE formation selectively recruits mTOR while excluding Vph1, thus clearly demonstrating that the puncta formation is not due to stress-induced aggregation.

The reviewer is right that conditions in our time-lapse experiments are not ideal for yeast growth, as cells are attached to the bottom of glass-bottom dishes, and there is

no shaking during imaging. The induced Tor1 puncta are however not protein aggregates, because they contain membrane as evidenced by FM4-64 staining and Atg22 (Figure 4A and 5B). In addition, we now show that Vph1 is excluded from the induced puncta (Figure 3F), further confirming their identity as genuine signaling endosomes.

Lastly, this paper provides evidence that signaling endosomes are formed from the vacuole rather than at a branching point in the endocytic pathway. Since this model differs from previous models, the authors should emphasize and discuss these differences to provide greater clarity to the field. Previous studies reported the requirement of HOPS and ESCRT for SEs. How does this fit into the current model?

This is indeed an important point. Our vacuole-origin model does not contradict the endosome-origin model previously proposed by the Ungermann group. In fact, we speculate that signaling endosomes form through the fusion of two precursor membranes, one derived from vacuoles (discovered in this work) and the other derived from endosomes (proposed by the Ungermann group). This fusion scenario would beautifully explain the “endosome-vacuole hybrid” protein/lipid composition of signaling endosomes. Our new data that plasma membrane transporter Dip5 passes through signaling endosomes (Figure S5D) further supports the endocytic pathway being another membrane source. We described the fusion scenario in the Discussion section of the original manuscript already, but for better clarification, we have now elaborated on this argument and added a new diagram (Figure S2G).

Our vacuole-origin model perfectly explains the requirement of HOPS and ESCRT. The lack of HOPS or ESCRT must cause a massive alteration in the composition of the vacuolar membrane, by compromising delivery of cargo proteins and lipids. This alteration could affect the formation of signaling endosomes from the vacuolar membrane, probably for many reasons. It might be however difficult to pinpoint one or two critical factors, given the broad impact of HOPS and ESCRT on the composition of the vacuolar membrane.

Minor Notes:

Fig 1b: The authors should provide one or two sentences explaining why a rapamycin survival assay was used to determine the functionality of mNG-Tor1.

This is a standard assay that was used to assess the functionality of, for example, GFP-fusion Tor1 constructs (PMID: 25046117). Nevertheless, we now further confirmed the functionality of mNG-Tor1 by assessing the phosphorylation of the TORC1 substrate Sch9 (Figure S1B).

Fig 3b: The SEs from cells exposed to 10 ng/ml aTc are significantly larger compared to those exposed to 1 ng/ml. Phosphomimetic fab1-6D (Fig 5c) also seems to have larger SEs. What could be the reason for the size differences observed?

We have now analyzed SE size. As shown in Figure S2H, there was no statistically meaningful difference in the puncta size between 1 ng/ml and 10 ng/ml aTc samples.

However, the reviewer is right that *fab1-6D* cells have bigger puncta (Figure S2I). This raises an interesting possibility that the size of signaling endosomes is fine-tuned by the local concentration of PI(3,5)P₂ there (which is higher in *fab1-6D* cells; see PMID: 33157024). An alternative interpretation could be that the bigger puncta in *fab1-6D* cells reflect a higher abundance of the TORC1 molecules at signaling endosomes (which can be expected as PI(3,5)P₂ anchors TORC1 to the membrane: PMID: 24478451).

Additional quantification and distribution analysis on SEs should be performed. In addition to assessing how many cells form puncta, how many puncta per cell are there?

We have done this now (Figure S2A to S2D). Most cells have only one signaling endosome, as previously noted (PMID: 33157024). We now think that this singularity of signaling endosome formation could be explained by the TORC1-Fab1-Atg18 positive feedback loop we discovered here. We have added a detailed argument in the Discussion (the second last paragraph).

Are certain cells more likely to form SEs compared to others?

Yes- as already mentioned, we now found a correlation with the cell cycle progression (Figure S4C and S4D). However, this does not fully explain the heterogeneity among a cell population, because SE formation was not 100% for any budding stage. We now discuss two possibilities; SE formation may be affected by (1) cellular replication age or (2) fluctuation of metabolic states (e.g., TORC1 activity) at a single cell level (the third last paragraph of Discussion). We plan to address these possibilities in future studies.

Is SE formation a one-time event, or does it occur multiple times within the same cell?

This is another exciting future direction. Answering this question would require longer time-lapse imaging, carried out over several generations in a physiological setting. We thus have started collaborating with our local colleague Claudiu Vasile Giuraniuc, who is expertized in yeast culture in a microfluidic device (PMID: 37750683). We are hopeful that this new approach will elucidate how the life cycle of signaling endosomes relates to cell cycle progression and cellular aging.

Reviewer #2 (Comments to the Authors (Required)):

Kenji Muneshige and Riko Hatakeyama provide a nice study to show that Atg18, in concert with retromer, is required for the formation of signaling endosomes (SE). SEs are structures containing Torc1 and its regulators and the full function of these compartments and how they are generated are important to resolve. These investigators made a system to induce Atg18 and watch the formation of SEs. The process is slow (~60-70 min) but helpful enough to show that FM4-64 that had been previously chased to the vacuole can be found in the newly formed SEs.

We thank the reviewer for the enthusiastic comments and insightful suggestions.

The authors are careful not to say that all SE membranes are derived from the vacuole, which is an important clarification. Though an explicit discussion of this point (that 'forward' traffic in the endosomal system may intersect in this compartment since these experiments do not rule that out) would be helpful for the reader.

We note the Reviewer's appreciation of our effort not to mislead readers by excluding involvement of "forward" traffic from canonical endosomes. We indeed believe this takes place, and now provide experimental confirmation by showing Dip5 localization to signaling endosomes (Figure S5D).

The merger of forward and reverse traffic would explain the "endosome-vacuole hybrid" composition of signaling endosomes. In the revised manuscript, we explicitly mentioned the forward traffic in the Discussion. Moreover, we emphasized the forward-reverse intersection/fusion model by depicting it in the new Figure S2G.

In general, the studies are nicely done and the main point substantiated. What distinguishes the SE as a unique endosome vs an outcropping of the vacuole with some Vps21/Rab5 on it is a matter of language perhaps, but gets at the heart of how to interpret the novelty of the central finding of this paper.

This is a good point. The available evidence favours the view that signaling endosomes as a separate organelle. In particular:-

1. As we discovered in this study, signaling endosomes are generated by the CROP complex, which cuts the vacuolar membrane.
2. Signaling endosomes appear highly protruded from the vacuolar surface, as confirmed by our new Airyscan images in Figure 3F, enlarged panels.
3. We show new data that FM4-64 is not efficiently exchanged between the vacuoles and mature signaling endosomes (Figure 4B), suggesting disconnection of their membranes.
4. We now show that the Dip5 transporter localizes to signaling endosomes (Figure S4D). This finding is difficult to reconcile with the vacuolar domain idea, as Dip5 is believed not to reach the vacuolar membrane.
5. In a similar vein, the Ungermann group demonstrated that the endocytosed alpha-factor reaches the lumen of signaling endosomes before it reaches the vacuolar lumen (PMID: 35404387). Alpha-factor did not diffuse from signaling

endosomes to the vacuole, suggesting the lumens of the two organelles are not connected.

Even with all this evidence, we recognize the importance of directly visualizing the interface of signaling endosomes and vacuoles, and have tried to observe signaling endosomes with electron microscopy. The main challenge is the scarcity of signaling endosomes, which are not formed in all cells, and when they are, there is usually only one signaling endosome per cell (Figure S2A). We therefore need to label signaling endosomes. Immuno-gold labelling of Tor1 was however unsuccessful, presumably due to the low expression level of Tor1. We have now switched our strategy to correlative light-electron microscopy (CLEM), which we are still optimizing. While we hope this CLEM approach will be fruitful, obtaining publishable images is not feasible within a reasonable revision timescale.

As is, I think the work is valuable and may be instructive for JCB readers.

Thank you for the positive assessment.

Here is a list of improvements that should be considered.

- The time course of Atg18 induction is likely slow based on the previously published description of this tet-controlled promoter construct. This study needs to have a time course of Atg18 induction at the various levels of tet and not just a single time point of various levels of tet.

We have now done this experiment (Figure S4A), which demonstrates time and dose-dependency as expected.

- A point the authors make is why SEs are only in a subset of cells. This is an intriguing point and observed in these Atg18-induction experiments and in normal cultures of WT cells. For this particular experiment, however, the finding would be greatly strengthened from showing induction of the functional mCherry-Atg18 protein with simultaneous imaging of a couple of the SE GFP markers to view throughout the population of cells to determine heterogeneity,

We confirmed the functionality of mCherry-Atg18 for SE formation, vacuolar morphology, and autophagy activity (Figure S1C to S1E). We then performed the suggested population analysis (Figure S4B). Interestingly, the heterogeneity in SE formation is not fully explained by heterogeneity of expression or puncta formation of Atg18 (i.e., there existed Tor1 puncta-positive and Atg18 puncta-negative cells and vice versa).

As a partial explanation, we identified a correlation between SE formation and cellular budding status, and thus cell cycle progression (Figure S4C and S4D). However, because the cell cycle still does not fully explain the heterogeneity, we have expanded the Discussion (the third last paragraph).

- Figure 4 shows the turn-on of Atg18. There is no quantitation of the number of compartments formed and the arrows actually look a bit arbitrary. For instance, you

could draw arrows at different structures at the 0-30 time points that are probably bunched vacuolar membrane giving the illusion of an SE. This kinetic analysis needs multiple samples and a set criteria to define when there is an SE vs a blobby vacuole.

We now track Vph1 in the time-lapse experiment (Figure 3F). Absence of Vph1 confirms structures as signaling endosomes, not small vacuoles. Kinetic analysis is shown in Figure 3B, which has now been improved by having n=3 and statistics.

- Figure 4 needs better microscopy. Deconvoluted Z-stacks (or even maximum point projections) would be helpful - this would show dots through out the cell. The Zeiss LSM 880 used for these studies may even have Airyscan to help resolution.

Those images are actually z-projections (apologies for the confusion, this is now clarified in Material and Methods). The main problem is the low expression level of Tor1, combined with the need to keep the laser power as weak as possible to prevent photo-bleaching and photo-toxicity. Even so, the present image quality is sufficient to confirm membrane supply from vacuoles.

Nevertheless, we followed the Reviewer's suggestion to perform Airyscan microscopy, and were pleased to obtain beautiful images of signaling endosomes (Figure 3F, see enlarged panels). Moreover, this analysis unexpectedly revealed a distinct vacuolar microdomain accommodating Tor1, likely stimulating further studies.

- Atg18 production is probably fairly slow here, preventing more acute experiments such as finding whether retromer substrate membrane proteins pass through this compartment on their way back to the TGN. Yet, even with slow Atg18 production, there is a missed opportunity to examine the order of recruitment to the forming SE. Things like Pib2, PI3,5, Torc1, Vps21, etc, may arrive in a different order. Provided there is a quantitative assay that considers several samples that can smooth the estimate of arrival times, this would really strengthen this type of biogenesis story. This could be informative and surprising - for instance, in the one set of micrographs provided, Atg22 seemingly arrives in concentrated areas before Tor1 (Fig 4).

We agree, it would be great if we could determine the order of events. However, these experiments are not technically trivial. The timing of signaling endosome formation is heterogeneous within a cell population (see the large error bars in Fig 3B), making it impossible to detect small timing differences between separate samples. Another possibility would be to simultaneously track multiple proteins in timelapse experiments, but in that case, we have to optimize the imaging settings for all the proteins, minimizing photo-toxicity and photo-bleaching. This is not easy, as components of the TORC1 pathway are expressed at a low level (which is why we made and used mNeonGreen-Tor1 instead of GFP-Tor1). In fact, we spent several months optimizing mNG-Tor1 time-lapse imaging. Unfortunately we cannot perform the extended set of experiments suggested with a timescale feasible for Kenji Muneshige, the PhD student first author.

- There is a lot of speculation about what might be special about Atg22 and its distribution in SE. This looks like the only protein besides Vph1 that was looked at, so speculation in this regard warrants inspection of a larger set of proteins - especially those that one might expect to recycle from vacuolar membranes at some point.

Thank you for an excellent suggestion. We now looked at an additional 16 proteins, among which 5 showed localization to signaling endosomes (Figure 5D). This list confirms the selective nature of SE targeting, and moreover, significantly expands our knowledge of the composition (and potential biological functions) of signaling endosomes.

- The authors state: "This membrane source for signaling endosomes differs starkly from our current knowledge of canonical endosomes, for which the membrane is supplied mainly from the plasma membrane and the Golgi apparatus via intracellular vesicle trafficking." There are no data in this manuscript showing that this does not happen for SEs - there is no test of 'forward' trafficking. Moreover, the idea that there is protein and lipid recycling from late endosomes and lysosomes back to early endosomal/TGN compartments - even routes dependent on Atg18 - have been articulated before so this claim is a bit exaggerated.

The reviewer is right. We modified the first paragraph of the Discussion to make it more accurate and balanced.

- The authors state: "This unique feature suggests that signaling endosomes should be most correctly viewed as a distinct organelle rather than a mere subpopulation of endosomes." This is helpful. Yet one can also consider the SE as a subdomain of the vacuolar compartment, leaving its communication with FM4-64 labelled vacuoles more to be expected.

Please see my earlier comments (numbered list in the first page of our response to Reviewer 2) on why we consider signaling endosomes as a distinct organelle.

There is also the possibility (just from the ATG22 result) that SEs serve as a signaling center and also as a sorting station, similar to ER exit sites where lipids and proteins such as Atg22 accumulate for carrier packaging.

The idea that signaling endosomes serve as a sorting station is indeed attractive (though speculative at this point). We have augmented our discussion of this point, and now further emphasize it with a new diagram (Figure S2G).

- Fix the reference formatting (relevant ones within a single set of '()').

Thank you, now amended accordingly.

Reviewer #3 (Comments to the Authors (Required)):

The manuscript by Muneshige and Hatakeyama represents an exciting advance in our understanding of signaling endosomes and TORC1 regulation. Many of the experimental approaches used in this paper are compelling, including the development and implementation of a new system to monitor signaling endosome biogenesis and the use of the tunable expression system for Atg18. The manuscript is clear and concise in its delivery, making the paper a joy to read. The novel finding that signaling endosomes arise from vacuoles in a manner dependent upon Atg18 and retromer is not only thought provoking but to my knowledge represents a new paradigm for regulation of membrane proteins at the vacuole. I am highly enthusiastic about this manuscript and overall, the quality of the data are very good.

We thank the reviewer for the enthusiastic and encouraging comments.

However, I have a few points for the authors to consider that may help bolster their claims.

Major Points:

1. Atg18 is required for signaling endosome formation but does not disrupt other endosomal compartments: This is the main takeaway from Figure 1, which is nicely presented. However, given the interplay between Golgi and endosomal compartments, it might be wise to also examine the distribution of a Golgi marked compartment (i.e., Sec7) to ensure it is unaltered in the *atg18Δ* cells.

We now have done this experiment (Figure 1I), which demonstrates no effect on Golgi as expected.

In addition, in Figure 1J, it would be useful to define the % of puncta co-localizing between mNG-Tor1 and mCherry-Atg18 (Pearsons and Manders coefficients).

This data now includes the requested quantification and a Venn diagram. Pearson coefficient etc are not suitable here, because Tor1 and Atg18 perfectly overlap on the vacuolar membrane, causing a false-high correlation.

Finally, in Figure 1J the vacuoles appear somewhat enlarged which raises the question on how the function of the mCherry-tagged Atg18 was assessed. It would be wonderful if the data demonstrating that mCherry-Atg18 is functional was included as a supplement.

Thank you for spotting this problem. We re-analyzed the images and realized that, as the reviewer points out, vacuoles were enlarged. We suspected this could be due to a high expression level of Atg18, because, in the original experiment, mCherry-Atg18 was overexpressed under the TEF promoter. Overexpression was needed because mCherry-Atg18 did not give a readily detectable signal when expressed under the native promoter.

We now tried milder overexpression using the NOP1 promoter. This new mCherry-ATG18 strain showed normal vacuolar morphology (Figure S1D). We also confirmed that the two other functions of Atg18, signaling endosome formation and autophagy activity, were normal in this new mCherry-Atg18 strain (Figure S1C and S1E). We thus repeated and confirmed colocalization using this new strain (Figure 1J).

2. Ivy1 continues to form puncta in the absence of Atg18: Some additional comment on why this is the case is warranted in the text of the manuscript. Does this suggest that Ivy1 might be a precursor to Atg18 recruitment during formation of signaling endosomes?

Exactly, we think Ivy1-positive endosomes are likely to be precursors of signaling endosomes. We have now elaborated this point by (i) showing that Ivy1-positive structures represent a pool of endosomes (Figure S2F) and (ii) discussing this point explicitly alongside a new illustrative cartoon (Figure S2G).

What is known about how Ivy1 is recruited to vacuole membranes? The fact that the authors indicate that the mNG-Ivy1 specifically may be localizing to subpopulations of endosomes raises the question of how the mNG-Ivy1 localization compares to other tagged forms of Ivy1. Please address these issues in the text; Past citations may suffice to clarify what is occurring.

Ivy1 localizes to vacuoles by binding to phospholipids and the Ypt7 Rab GTPase (PMID: 25999476).

Our observations are not specific to the mNG tag: we noticed that Christian Ungermann's group reached the same conclusion that Ivy1 also localizes to other endosomal subpopulations (PMID: 35404387—see detailed arguments in this paper's published Review History). We have now cited this paper in this context.

3. Atg18 membrane scission function is required for formation of Tor1-containing signaling endosomes: In Figure 2A, only Tor1 puncta are monitored in the *hsv2* and *atg21* cells. It would be useful here to look at other signaling endosome markers to make the finding more compelling.

We have now looked at Gtr1 and Pib2 markers, and we show the results in Figures S3A and S3D to provide additional evidence.

In Figure 2F, the T56E and FGGG variants of 3FLAG-Atg18 are not expressed as well as the WT or Sloop mutants. The statement on line 166 of page 5 should be revised to reflect that difference; while these variants were expressed, they were expressed at a lower level than WT Atg18 (2-3 fold lower? Consider quantifying).

The reviewer is right. We now have quantification of Figure 1F under the blot. The text has been modified accordingly ("*these Atg18 mutants (which were all expressed robustly though at varied levels*").

These Atg18 mutants are reported to impact membrane scission, but the Atg18 lipid transfer activities, as part of Atg2-9-18 complex, should be intact. Perhaps assessing this facet of Atg18 function would be beneficial here (or citing where this has been shown previously).

We now cite the corresponding papers that assessed autophagy activity of those Atg18 mutants (De Leo et al., 2021; Courtellemont et al., 2022; Gopaldass and Mayer, 2024).

Similar to my comment for the mCherry-Atg18, what assays - other than the formation of the mNG-Tor1 puncta -- were done to ensure that the FLAG tagged Atg18 employed was functional? I suggest that monitoring vacuole morphology and localization/abundance of Gtr1 and/or Pib2 puncta would be helpful orthogonal approaches that could bolster Figure 2.

As suggested, we now show Gtr1 and Pib2 puncta formation (Figure S1F and S1G) and vacuolar morphology (Figure S1H). In addition, we checked autophagy activity (Figure S1I). All these data demonstrate the functionality of our 3FLAG-Atg18 construct.

4. Atg18 is sufficient to allow for Tor1 puncta formation in add back experiments: These are exciting findings that are foundational to this study and so should be highly rigorous. In Figure 3B, the time course results for the % puncta formation has no error bars or statistical measures to ensure that the change from a few percent (perhaps increase the number of tick marks on your scale bar here please) to a maximum of 15% is statistically different. I realize that this requires considerable effort, but in my view it is required for the result to be compelling.

Figure 3B now has n=3 with quantification and statistics.

The absence of Vph1 in the mNG-Tor1 puncta and the co-localization of mCherry-Vps21 with mNG-Tor1 puncta in Figures 3D and 3E should have some kind of measurement added (i.e., pearsons and manders).

We now include quantification. Pearson coefficient etc is not suitable for analysis here, because Tor1 and Vph1 perfectly overlap on the vacuolar membrane, which causes a falsely high correlation.

5. Atg18 drives formation of a distinct, vacuole-derived, compartment: A critical aspect of the study will be demonstrating that the Atg18 puncta formed in WTC846 expression system are discrete from the vacuole membrane. To address this comment likely requires 3D imaging or EM. This is especially important as the claim is being made in the discussion that these are discrete organelles. The FM imaging nicely shows that the membrane protrusion from the vacuole co-localizes with Tor1, but it does not demonstrate that the Tor1 puncta is separate from the vacuole. From the data presented, it is hard to discern that the signaling endosomes marked by Tor1 are distinct compartments from the vacuole itself. Is there 3D imaging or an EM reference that can help distinguish between the possibilities that (i) the signaling endosome is a sub-

compartment of the vacuole or (ii) the signaling endosome is a γ free and distinct membrane bound compartment?

Please see my comments to Reviewer 2 (numbered list in the first page of our response to Reviewer 2) on why we consider signaling endosomes as a distinct organelle.

Minor points

A) In Figure 2D, the loss of retromer most likely disrupts endosome formation in general. Is the loss of retromer here having a direct or indirect effect on signaling endosome formation?

Answering this question would require a separation-of-function mutant of retromer subunits defective only in the CROP complex, which is to our knowledge not available. We now mentioned this caveat in the manuscript (Results section regarding Figure 2D “*the phenotype of retromer mutants can be caused by either the loss of the CROP complex or that of other retromer-dependent trafficking events,*”).

B) In Figure 4, there appear to be other concentrated signal points for Tor1 that do not go on to form as robust an accumulation. This made me wonder if, like endocytic sites, there is a critical concentration of factors that needs to be assembled on the vacuole to generate these signaling endosomes. γ Some mention of the smaller clusters of Tor1 signal might be useful.

This is an interesting idea, particularly in light of our finding of the positive feedback loop involving TORC1, Fab1, and Atg18. This feedback loop may amplify signals triggered by a small (possibly even spontaneous) fluctuation of these molecules, allowing such signals to pass a “critical concentration”. This model could explain the observed population heterogeneity of SE-positive and -negative cells. We have incorporated this argument into the Discussion (third- and second-last paragraphs).

In this context, our new data with Airyscan microscopy (Figure 3F, see enlarged panels) revealed many microdomain-like clusters of vacuolar TORC1. We indeed think these TORC1 clusters may seed signaling endosomes, and we now mention this point in the Discussion.

C) What is the fate of the signaling endosome? Once formed from the vacuole, where does it go? Is it destined for fusion with the PM like other vesicles formed from lysosomes/vacuoles? Does it fuse to other compartments or does it return to the vacuole at some point? Some comment on this in the discussion seems warranted.

We totally agree with the reviewer. The destination of signaling endosomes is as important as how they are formed. We are now trying to monitor signaling endosomes over long periods using a microfluidic device, which we hope will hint at the terminal fate of signaling endosomes. However, the focus of the present paper is on the biogenesis of signaling endosomes. We mention investigation of their fate as a future direction (last paragraph of Discussion).

D) Text suggested changes -

Line 50 on page 3 - Signaling endosomes accommodate - consider revising to accumulate

Changed, thank you.

Lines 53 and 56 on page 3 - no need to have 'its' in either of these sentences

Deleted, thank you.

Line 292-293 on page 8 - Biogenesis is repeated.

Fixed, thank you for spotting this.

**TORC1-containing signaling endosomes source membrane**
**from vacuoles**

Kenji Muneshige and Riko Hatakeyama*

Institute of Medical Sciences, School of Medicine, Medical Sciences & Nutrition, University of
Aberdeen

*Corresponding author;

ORCID 0000-0002-3065-6639

riko.hatakeyama@abdn.ac.uk

Institute of Medical Sciences, University of Aberdeen, Foresterhill, Aberdeen AB25 2ZD, UK

**Running title**

Signaling endosomes originate from vacuoles

**Keywords**

yeast; signaling endosome; vacuole; Target of Rapamycin Complex 1 (TORC1); Atg18

**Abbreviations**

TORC1: Target of Rapamycin Complex 1

CROP: Cutting Retromer-on-PROPPIN

PI(3,5)P₂: phosphatidylinositol 3,5-bisphosphate

PI3P: phosphatidylinositol 3-phosphate

**Summary**

Membrane is supplied by vacuoles during biogenesis of yeast signaling endosomes, in a process
mediated by the membrane-cutting CROP complex and promoted by TORC1 kinase activity.

**Abstract**

[revised manuscript text omitted]

subcellular loci (~~presumably other subpopulations of endosomes~~) as well. Consistently, the lvy1
puncta were reduced but not absent from *atg18Δ* cells (Figure 1FG). Most lvy1 puncta were positive
for Vps21 (Figure S2E and S2F) in agreement with a previous study (Gao et al., 2022), suggesting that
they are endosomes (Figure S2G; see also Discussion).

Formatted: Not Highlight

To confirm that Atg18 is not required for the formation of other endosomal subpopulations, we
examined the localization of ~~Vps21 and Vps4~~ and Vps21, endosomal markers not specific to signaling
endosomes (Hatakeyama et al., 2019; Chen et al., 2021). These proteins indeed retained their
punctal localization pattern in *atg18Δ* cells (Figure 1GH and 1H), suggesting that the requirement
of Atg18 is specific to signaling endosomes. We also confirmed that the trans-Golgi network marked
by Sec7 is unaffected (Figure 1I).

Consistent with the direct involvement of Atg18 in signaling endosome formation and/or function,
mCherry-Atg18 formed perivacuolar puncta partially overlapping with Tor1 puncta (Figure 1J). Our
mCherry-Atg18 was functional for signaling endosome formation, vacuolar fission, and autophagy as
shown in Figure S1C to S1E. Collectively, our observations suggest an essential role for Atg18 in
signaling endosome formation. We found that mCherry-Atg18 localizes to signaling endosomes
(Figure 1J), consistent with direct involvement of Atg18 in signaling endosome formation and/or
function.

**The Atg18-containing CROP complex mediates signaling endosome biogenesis**

Having established the requirement of Atg18 for signaling endosome formation, we addressed the
underlying molecular mechanisms. We first tested whether this function of Atg18 is shared by other
structurally similar PROPPINs ~~Atg21 and Hsv2~~ and ~~Atg21~~ (Krick et al., 2008). Signaling endosomes
were observed at normal levels in ~~hsv2Δ and atg21Δ~~ and hsv2Δ cells (Figure 2A, S3A and S3B),
suggesting that this function is unique to Atg18 among the three yeast PROPPINs.

Atg18 functions in the context of distinct protein complexes (Figure 2B). The Atg18-Atg2-Atg9
trimeric complex promotes the expansion of the pre-autophagosomal membrane by transferring
lipid molecules from the endoplasmic reticulum (Maeda et al., 2019; Valverde et al., 2019;
Gómez-Sánchez et al., 2018). More recently, the Mayer and Thumm groups independently reported
another Atg18-containing complex, the CROP (Cutting Retromer-on-PROPPIN) complex
(Courtellemont et al., 2022; Marquardt et al., 2023; Gopaldass and Mayer, 2024; Marquardt and
Thumm, 2023). CROP is comprised of Atg18, Vps26, Vps29, and Vps35, the latter three of which are
subunits of the retromer sorting complex (Seaman et al., 1998). CROP cuts the membrane of endo-
lysosomal compartments, promoting formation of tubulo-vesicular transport carriers from
endosomes, and vacuolar fission and fragmentation (De Leo et al., 2021; Gopaldass et al., 2017;
Courtellemont et al., 2022; Marquardt et al., 2023).

We addressed the protein complex and functions through which Atg18 supports signaling endosome
formation by deleting each binding partner. Signaling endosomes were intact in *atg2Δ* and *atg9Δ*
cells (Figure 2C), ruling out the involvement of the Atg18-Atg2-Atg9 complex. In contrast, *vps26Δ*,
*vps29Δ* and *vps35Δ* mutants had significantly fewer signaling endosomes than wild-type cells (Figure

2D, S3C and S3D), suggesting a role for CROP-mediated membrane fission in signaling endosome
formation. ~~Because the phenotype of retromer mutants can be caused by either the loss of the~~
~~CROP complex or that of other retromer-dependent trafficking events, we next examined Atg18~~
~~mutants defective in the CROP function. To corroborate this notion, we~~ analyzed three Atg18
mutants specifically defective in CROP complex formation and/or membrane fission, T56E, SLoop,
and FG3G mutants (Figure 2E) (De Leo et al., 2021; Courtellemont et al., 2022; Gopaldass and Mayer,
2024). As expected, these Atg18 mutants (which were all expressed ~~well robustly though at varied~~
~~levels~~; Figure 2F top) failed to support signaling endosome formation (Figure 2F bottom). The 3FLAG-
Atg18 construct used here was functional for signaling endosome formation, vacuolar fission, and
autophagy as shown in Figure S1F to S1I.

Our results suggest a major role for the CROP complex in signaling endosome formation.
Consistently, retromer subunits localized to signaling endosomes along with other endosomal
subpopulations (Figure 2G, S3E to S3G). We nevertheless noticed that the defects in retromer
mutants (Figure 2D) are significant but only partial, differing from the complete loss of signaling
endosomes in *atg18Δ* cells. This observation aligns well with the fact that Atg18 can act alone to cut
a membrane, albeit less efficiently than when it is in the CROP complex (Gopaldass et al., 2017;
Courtellemont et al., 2022).

**Development of an inducible system to study signaling endosome biogenesis**

Formatted: Font: Bold

The participation of the CROP complex indicates that signaling endosome biogenesis involves a
membrane fission process. Because 1) the CROP complex cuts the vacuolar membrane, 2) signaling
endosomes are always adjacent to vacuoles, and 3) the protein and lipid composition significantly
overlap between signaling endosomes and vacuoles, it was plausible that vacuoles supply the
membrane, through a fission process, to newly born signaling endosomes.

We sought to test this hypothesis by monitoring signaling endosome formation in real-time. To this
end, we took advantage of the complete loss of signaling endosomes in *atg18Δ* cells, identifying
*ATG18* as Gene X (Figure 1A). To conditionally express Atg18, we utilized the WTC_{g46} toolkit, an
improved tetracycline-inducible gene expression system (Azizoglu et al., 2021). We confirmed that
anhydrotetracycline induces 3FLAG-Atg18 expression in a dose-dependent manner in the WTC_{g46}
*ATG18* strain (Figure 3A and S4A). Without anhydrotetracycline treatment, Atg18 expression was
undetectable. The Atg18 expression level with 1 ng/ml anhydrotetracycline approximately
corresponded to the endogenous level (Figure 3A first lane), so we used this concentration of
anhydrotetracycline in the following experiments unless specified otherwise. The expression level of
Tor1 did not change after anhydrotetracycline treatment, meaning that anhydrotetracycline-
dependent changes in Tor1 distribution described below are not due to changes in the Tor1
abundance.

Formatted: Subscript

Formatted: Subscript

As expected, WTC_{g46} -*ATG18* cells showed no Tor1 puncta without anhydrotetracycline treatment
(Figure 3B). Tor1 puncta started to appear around 30 to 60 minutes after the addition of
anhydrotetracycline (Figure 3B and S2H). ~~The number of Tor1 puncta correlated with the~~
~~concentration of anhydrotetracycline, i.e., the expression level of Atg18. Interestingly however,
[revised manuscript text omitted]

Tukey's multiple comparisons were used.

Supplementary materials

Figure S1. Functionality of tagged protein constructs.

Figure S2. Extended analysis of signaling endosomes.

Figure S3. Further evidence of CROP-mediated signaling endosome biogenesis.

Figure S4. Additional data from Atg18 induction experiments.

Figure S5. Search for additional signaling endosomal proteins.

Table S1. Yeast strains used in this study.

Table S2. Plasmids used in this study.

Formatted: Font: Bold

Formatted: Font: Not Bold

Formatted: Font: Not Bold

Formatted: Font: Not Bold

Formatted: Font: Not Bold

Formatted: Font: Not Bold

**Acknowledgments**

We thank Claudio De Virgilio (University of Fribourg) and Christian Ungermann (University of
Osnabrück-University) for yeast strains, plasmids, and for helpful suggestions; Takahiro Shintani
(Tohoku University) for plasmids and helpful suggestions; Maya Schuldiner (Weizmann Institute of
Science) and Michael Knop (Heidelberg University) for yeast strain libraries; Takeshi Noda (Osaka
University), Satoshi Okada (Kyushu University, NBRP plasmid BYP9806), and Joerg Stelling (ETH
Zurich, Addgene plasmid FRP2350/2365) for plasmids; Tatsuya Maeda (Hamamatsu University School
of Medicine) for the anti-Tor1 antibody; Daniel Paterson, -Patryk Marcinkowski, Eri Hirata, and Saran
Babooraj (previously University of Aberdeen) for their contribution to the visual screening using
YeastRGB; members of the Hatakeyama lab and Chromosome & Cellular Dynamics Section
(University of Aberdeen) and Tokai TOR Conference for discussion; Megan Robertson, Colin Ferguson,
Arrosan Rajalingam, and Microscopy and Histology Core Facility members (University of Aberdeen)
for technical support; Anne Donaldson (University of Aberdeen) for advising on the manuscript. This
work was funded by BBSRC (BB/V016334/1 and BB/X018229/1 to RH) and the University of
Aberdeen. KM is a recipient of the JASSO scholarship. The authors declare no conflict of interest.

**Author contributions**

Conceptualization: KM and RH; Methodology: KM; Investigation: KM; Formal Analysis: KM;
Validation: KM; Supervision: RH; Funding acquisition: RH; Visualization: RH; Writing – original draft:
KM and RH; Writing – review & editing: RH.

[revised manuscript text omitted]

666 L.X. Heinz, C. Kraft, K.L. Bennett, C. Indiveri, L.A. Huber, and G. Superti-Furga. 2015. SLC38A9 is a
667 component of the lysosomal amino acid sensing machinery that controls mTORC1. *Nature*.
519:477–81. doi:10.1038/nature14107.

Scorrano, L., M.A. De Matteis, S. Emr, F. Giordano, G. Hajnóczky, B. Kornmann, L.L. Lackner, T.P.
Levine, L. Pellegrini, K. Reinisch, R. Rizzuto, T. Simmen, H. Stenmark, C. Ungermann, and M.
Schuldiner. 2019. Coming together to define membrane contact sites. *Nat Commun*. 10.
doi:10.1038/s41467-019-09253-3.

Seaman, M.N.J., J.M. Mccaffery, and S.D. Emr. 1998. A Membrane Coat Complex Essential for
Endosome-to-Golgi Retrograde Transport in Yeast. 142. 665–681 pp.

Shimobayashi, M., and M.N. Hall. 2014. Making new contacts: The mTOR network in metabolism and
signalling crosstalk. *Nat Rev Mol Cell Biol*. 15:155–162. doi:10.1038/nrm3757.

Sikorski, R.S., and P. Hieter. 1989. A System of Shuttle Vectors and Yeast Host Strains Designed for
Efficient Manipulation of DNA in *Saccharomyces cerevisiae*.

Stringer, C., T. Wang, M. Michaelos, and M. Pachitariu. 2021. Cellpose: a generalist algorithm for
cellular segmentation. *Nat Methods*. 18:100–106. doi:10.1038/s41592-020-01018-x.

Sturgill, T.W., A. Cohen, M. Diefenbacher, M. Trautwein, D.E. Martin, and M.N. Hall. 2008. TOR1 and
TOR2 have distinct locations in live cells. *Eukaryot Cell*. 7:1819–1830. doi:10.1128/EC.00088-08.

Suzuki, S.W., and S.D. Emr. 2018a. Membrane protein recycling from the vacuole/lysosome
membrane. *Journal of Cell Biology*. 217:1623–1632. doi:10.1083/jcb.201709162.

Suzuki, S.W., and S.D. Emr. 2018b. Retrograde trafficking from the vacuole/lysosome membrane.
*Autophagy*. 14:1654–1655. doi:10.1080/15548627.2018.1496719.

Toulmay, A., and W.A. Prinz. 2013. Direct imaging reveals stable, micrometer-scale lipid domains that
segregate proteins in live cells. *Journal of Cell Biology*. 202:35–44. doi:10.1083/jcb.201301039.

Valverde, D.P., S. Yu, V. Boggavarapu, N. Kumar, J.A. Lees, T. Walz, K.M. Reinisch, and T.J. Melia. 2019.
ATG2 transports lipids to promote autophagosome biogenesis. *Journal of Cell Biology*.
218:1787–1798. doi:10.1083/JCB.201811139.

Vida, T.A., and S.D. Emr. 1995. A new vital stain for visualizing vacuolar membrane dynamics and
endocytosis in yeast. *J Cell Biol*. 128:779–792. doi:10.1083/jcb.128.5.779.

Wang, S., Z.-Y. Tsun, R.L. Wolfson, K. Shen, G.A. Wyant, M.E. Plovanich, E.D. Yuan, T.D. Jones, L.
Chantranupong, W. Comb, T. Wang, L. Bar-Peled, R. Zoncu, C. Straub, C. Kim, J. Park, B.L.

[revised manuscript text omitted]

Formatted: Font: Not Bold, Font color: Text 1

895 **Tables**

896 **Table S1. Yeast strains used in this study**

ID	Genotype
YL516	[BY4741/2] MATa; his3Δ1-leu2Δ0-ura3Δ0
yRL795	[YL516] mNeonGreen-TOR1
MP1632	[YL516] tor1Δ::kanMX
yRL897	[yRL795] atg18Δ::kanMX
RKH94	[YL516] GFP-GTR1
yRL392	[RKH94] atg18Δ::kanMX
RKH158	[YL516] PIB2²⁰⁰-EGFP
yRL399	[RKH158] atg18Δ::kanMX
RKH486	[YL516] his3Δ1::GFP-SCH9¹⁻¹⁸²::SpHIS5
yRL402	[RKH486] atg18Δ::kanMX
yRL803	[YL516] IVY1-mNeonGreen::HIS3
yRL1230	[yRL803] atg18Δ::kanMX
yRL1170	[YL516] URA3::P_{NOP1}-GFP-VPS21
yRL1201	[yRL1170] atg18Δ::kanMX
yRL1173	[YL516] VPS4-mNeonGreen::hphNT1
yRL1233	[yRL1173] atg18Δ::kanMX
yRL1105	[yRL795] natNT2::P_{TEF2}-mCherry-ATG18
yRL1058	[yRL795] atg21Δ::kanMX
yRL1060	[yRL795] hsv2Δ::kanMX
yRL1053	[yRL795] atg2Δ::kanMX
yRL1055	[yRL795] atg9Δ::kanMX
yRL1063	[yRL795] vps26Δ::kanMX
yRL1066	[yRL795] vps29Δ::kanMX

Source

Click or tap here to enter text.

This study

Click or tap here to enter text.

This study

Click or tap here to enter text.

This study

Click or tap here to enter text.

This study

Click or tap here to enter text.

This study

This study

This study

This study

This study

This study

This study

This study

This study

This study

This study

This study

This study

This study

Formatted: Space After: 8 pt, Line spacing: Multiple 1.08 li

Formatted: Space After: 8 pt, Line spacing: Multiple 1.08 li

Formatted: Space After: 8 pt, Line spacing: Multiple 1.08 li

Formatted: Space After: 8 pt, Line spacing: Multiple 1.08 li

Formatted: Space After: 8 pt, Line spacing: Multiple 1.08 li

Formatted: Space After: 8 pt, Line spacing: Multiple 1.08 li

Formatted: Space After: 8 pt, Line spacing: Multiple 1.08 li

Formatted: Space After: 8 pt, Line spacing: Multiple 1.08 li

Formatted: Space After: 8 pt, Line spacing: Multiple 1.08 li

Formatted: Space After: 8 pt, Line spacing: Multiple 1.08 li

Formatted: Space After: 8 pt, Line spacing: Multiple 1.08 li

Formatted: Space After: 8 pt, Line spacing: Multiple 1.08 li

Formatted: Space After: 8 pt, Line spacing: Multiple 1.08 li

Formatted: Space After: 8 pt, Line spacing: Multiple 1.08 li

Formatted: Space After: 8 pt, Line spacing: Multiple 1.08 li

Formatted: Space After: 8 pt, Line spacing: Multiple 1.08 li

Formatted: Space After: 8 pt, Line spacing: Multiple 1.08 li

Formatted: Space After: 8 pt, Line spacing: Multiple 1.08 li

Formatted: Space After: 8 pt, Line spacing: Multiple 1.08 li

Formatted: Space After: 8 pt, Line spacing: Multiple 1.08 li

Formatted: Space After: 8 pt, Line spacing: Multiple 1.08 li

Formatted: Space After: 8 pt, Line spacing: Multiple 1.08 li

Formatted: Space After: 8 pt, Line spacing: Multiple 1.08 li

Formatted: Space After: 8 pt, Line spacing: Multiple 1.08 li

| | | |
|---|---|---|
| ~~yRL1069~~ | ~~[yRL795] *vps35Δ::kanMX*~~ | ~~This study~~ |
| ~~yRL913~~ | ~~[yRL795] *ura3Δ0::$P_{RNR2}$-tetR-NLS-tup1-$P_{7tet.1}$-tetR-NLS-URA3 $P_{ATG18}$::hphNT1-$P_{7tet.1}$-3xFLAG-ATG18*~~ | ~~This study~~ |
| ~~yRL1190~~ | ~~[yRL795] *ura3Δ0::$P_{RNR2}$-tetR-NLS-tup1-$P_{7tet.1}$-tetR-NLS-URA3 3FLAG-ATG18*~~ | ~~This study~~ |
| ~~yRL910~~ | ~~[yRL795] *ura3Δ0::$P_{RNR2}$-tetR-NLS-tup1-$P_{7tet.1}$-tetR-NLS-URA3 $P_{ATG18}$::hphNT1-$P_{7tet.1}$-ATG18*~~ | ~~This study~~ |
| ~~yRL1153~~ | ~~[yRL910] *VPH1-mCherry-V5::kanMX*~~ | ~~This study~~ |
| ~~yRL1192~~ | ~~[yRL910] *natNT2::$P_{TEF2}$-mCherry-VPS21*~~ | ~~This study~~ |
| ~~yRL1194~~ | ~~[yRL795] *natNT2::$P_{NOP1}$-mCherry-ATG22*~~ | ~~This study~~ |
| ~~yRL1197~~ | ~~[yRL910] *natNT2::$P_{NOP1}$-mCherry-ATG22*~~ | ~~This study~~ |
| ~~yRL1270~~ | ~~[yRL910] *natNT2::$P_{GAL1}$-mCherry-ATG22*~~ | ~~This study~~ |
| ~~yRL1287~~ | ~~[BY4741] MATa; *his3Δ1 leu2Δ0 met15Δ0 mNeonGreen-TOR1 ura3Δ0::$P_{RNR2}$-tetR-NLS-tup1-$P_{7tet.1}$-tetR-NLS-URA3 $P_{ATG18}$::hphNT1-$P_{7tet.1}$-ATG18*~~ | ~~This study~~ |
| ~~yRL1299~~ | ~~[BY4741] MATa; *his3Δ1 leu2Δ0 met15Δ0 mNeonGreen-TOR1 ura3Δ0::$P_{RNR2}$-tetR-NLS-tup1-$P_{7tet.1}$-tetR-NLS-URA3 $P_{ATG18}$::hphNT1-$P_{7tet.1}$-ATG18 fab1$^{S202A/S203A/S204A/T206A/S208A/S210A}$*~~ | ~~This study~~ |
| ~~yRL1290~~ | ~~[BY4741] MATa; *his3Δ1 leu2Δ0 met15Δ0 mNeonGreen-TOR1 ura3Δ0::$P_{RNR2}$-tetR-NLS-tup1-$P_{7tet.1}$-tetR-NLS-URA3 $P_{ATG18}$::hphNT1-$P_{7tet.1}$-ATG18 fab1$^{S202D/S203D/S204D/T206D/S208D/S210D}$*~~ | ~~This study~~ |

~~**Table S2. Plasmids used in this study**~~

| ~~ID~~ | ~~Name~~ | ~~Description~~ | ~~Source~~ |
|---|---|---|---|
| ~~pRL127~~ | ~~pRS413~~ | ~~CEN/ARS, *HIS3*~~ | Click or tap here to enter te |
| ~~pRL129~~ | ~~pRS415~~ | ~~CEN/ARS, *LEU2*~~ | Click or tap here to enter te |
| ~~pRL130~~ | ~~pRS416~~ | ~~CEN/ARS, *URA3*~~ | Click or tap here to enter te |
| ~~pRL479~~ | ~~p1379~~ | ~~CEN/ARS, *MET15*~~ | Click or tap here to enter te |
| ~~pRL910~~ | ~~pML104i/gTOR1-N~~ | ~~2μ, *URA3, $P_{GAP}$-SpCas9, $P_{SNR52}$-gTOR1-N*~~ | Click or tap here to enter te |
| ~~pRL886~~ | ~~BYP9806~~ | ~~*CSE4-mNeonGreen-HIS3*~~ | Click or tap here to enter te |
| ~~pRL913~~ | ~~pRL913~~ | ~~*$P_{TOR1}$-mNeonGreen-linker-TOR1$^{1-426}$*~~ | ~~This study~~ |
| ~~pRL942~~ | ~~FRP2365~~ | ~~Integrative, *URA3, $P_{RNR2}$-tetR-NLS-tup1-$P_{7tet.1}$p-tetR-NLS*~~ | Click or tap here to enter te |

ϕRL940	FRP2350	hphNT1-P₇₀₁₋₁-3xFLAG	
ϕRL1007	ϕRS416-3FLAG-ATG18	CEN/ARS, URA3, 3xFLAG-ATG18	This study
ϕRL1022	ϕRS416-3FLAG-ATG18T56E	CEN/ARS, URA3, 3xFLAG-ATG18^{T56E}	This study
ϕRL1025	ϕRS416-3FLAG-ATG18SLoop	CEN/ARS, URA3, 3xFLAG-ATG18^{Y367K/D369M/K372Y/M375D}	This study
ϕRL1027	ϕRS416-3FLAG-ATG18FGGG	CEN/ARS, URA3, 3xFLAG-ATG18^{R285G/R286G}	This study

Click or tap here to enter text

Formatted: Space After: 8 pt, Line spacing: Multiple 1.08 li

Formatted: Space After: 8 pt, Line spacing: Multiple 1.08 li

Formatted: Space After: 8 pt, Line spacing: Multiple 1.08 li

Formatted: Space After: 8 pt, Line spacing: Multiple 1.08 li

Formatted: Space After: 8 pt, Line spacing: Multiple 1.08 li

Figure 1

**Figure 1. Atg18 is required for signaling endosome formation.**

(A) Strategy to induce and synchronize *de novo* formation of signaling endosomes (SE). See text for
details. (B) Functionality of mNeonGreen-Tor1 (mNG-Tor1) used in this study. Serial dilutions of the
indicated strains were spotted and grown on YPD agar plates containing rapamycin at the indicated
final concentration. (C) (i) Requirement of Atg18 for the formation of signaling endosomes but not
that of canonical endosomes. Wild-type and *atg18Δ* cells expressing the indicated fusion proteins
were analyzed by fluorescent microscopy. Arrowheads indicate signaling endosomes. Scale bars: 5
μ m. Quantifications of puncta-containing cells are shown on the right (mean \pm SEM, n=3, >100 cells
were analyzed for each biological replicate). In (H) and (I), because all the cells contained dots,
[revised manuscript text omitted]

February 3, 2025

RE: JCB Manuscript #202407021R

Riko Hatakeyama
University of Aberdeen

Dear Dr. Hatakeyama:

Thank you for submitting your revised manuscript entitled "TORC1-containing signaling endosomes source membrane from vacuoles". The reviewers all appreciate your careful revisions and responses and find your study much improved. While they have a few remaining issues, editorially we find that you have addressed all essential issues and that your study is suitable for publication in JCB pending addressing their final concerns with text edits as well as final revisions necessary to meet our formatting guidelines (see details below).

A. MANUSCRIPT ORGANIZATION AND FORMATTING:

1) Text limits: Character count for Articles is < 40,000, not including spaces. Count includes abstract, introduction, results, discussion, and acknowledgments. Count does not include title page, figure legends, materials and methods, references, tables, or supplemental legends.

2) Figures limits: Articles may have up to 10 main text figures.

3) Figure formatting: Scale bars must be present on all microscopy images, including inset magnifications. Molecular weight or nucleic acid size markers must be included on all gel electrophoresis. Aspect ratios of images may not be altered.

4) Statistical analysis: Error bars on graphic representations of numerical data must be clearly described in the figure legend. The number of independent data points (n) represented in a graph must be indicated in the legend. Statistical methods should be explained in full in the materials and methods. For figures presenting pooled data the statistical measure should be defined in the figure legends. Please also be sure to indicate the statistical tests used in each of your experiments (either in the figure legend itself or in a separate methods section) as well as the parameters of the test (for example, if you ran a t-test, please indicate if it was one- or two-sided, etc.). Also, if you used parametric tests, please indicate if the data distribution was tested for normality (and if so, how). If not, you must state something to the effect that "Data distribution was assumed to be normal but this was not formally tested."

5) Abstract and title: The abstract should be no longer than 160 words and should communicate the significance of the paper for a general audience. The title should be less than 100 characters including spaces. Make the title concise but accessible to a general readership.

* For clarity we suggest the following edited title: "Vacuoles provide the source membrane for TORC1-containing signaling endosomes"

6) Materials and methods: Should be comprehensive and not simply reference a previous publication for details on how an experiment was performed. Please provide full descriptions in the text for readers who may not have access to referenced manuscripts.

7) All antibodies, cell lines, animals, and tools used in the manuscript should be described in full, including accession numbers for materials available in a public repository such as the Resource Identification Portal. Please be sure to provide the sequences for all of your primers/oligos and RNAi constructs in the materials and methods. You must also indicate in the methods the source, species, and catalog numbers (where appropriate) for all of your antibodies. Please also indicate the acquisition and quantification methods for immunoblotting/western blots.

8) Microscope image acquisition: The following information must be provided about the acquisition and processing of images:

- a. Make and model of microscope
- b. Type, magnification, and numerical aperture of the objective lenses
- c. Temperature
- d. Imaging medium

- e. Fluorochromes
- f. Camera make and model
- g. Acquisition software
- h. Any software used for image processing subsequent to data acquisition. Please include details and types of operations involved (e.g., type of deconvolution, 3D reconstitutions, surface or volume rendering, gamma adjustments, etc.).

10) Supplemental materials: There are strict limits on the allowable amount of supplemental data. Articles may have up to 5 supplemental figures. Please also note that tables, like figures, should be provided as individual, editable files. A summary of all supplemental material should appear at the end of the Materials and methods section.

13) ORCID IDs: ORCID IDs are unique identifiers allowing researchers to create a record of their various scholarly contributions in a single place. Please note that ORCID IDs are now *required* for all authors. At resubmission of your final files, please be sure to provide your ORCID ID and those of all co-authors.

Please note that JCB now requires authors to submit Source Data used to generate figures containing gels and Western blots with all revised manuscripts. This Source Data consists of fully uncropped and unprocessed images for each gel/blot displayed in the main and supplemental figures. For assays performed using capillary electrophoresis and/or immunoassay-based detection, authors should instead provide the electropherogram graph(s) for each experiment, plotting fluorescence/chemiluminescence intensity vs. molecular weight/size. Since your paper includes cropped gel and/or blot images, please be sure to provide one Source Data file for each figure gels, blots, and/or capillary electrophoresis assays along with your revised manuscript files. File names for Source Data figures should be alphanumeric without any spaces or special characters (i.e., SourceDataF#, where F# refers to the associated main figure number or SourceDataFS# for those associated with Supplementary figures). For traditional gels and blots, the lanes of the gels/blots should be labeled as they are in the associated figure, the place where cropping was applied should be marked (with a box), and molecular weight/size standards should be labeled wherever possible. For capillary electrophoresis assays, each trace in the graph should be color-coded and labeled to indicate which protein, gene, or sample is being measured (please try to avoid red/green combinations to accommodate our color-blind readers).

Journal of Cell Biology now requires a data availability statement for all research article submissions. These statements will be published in the article directly above the Acknowledgments. The statement should address all data underlying the research presented in the manuscript. Please visit the JCB instructions for authors for guidelines and examples of statements at (<https://rupress.org/jcb/pages/editorial-policies#data-availability-statement>).

B. FINAL FILES:

-- High-resolution figure and MP4 video files: See our detailed guidelines for preparing your production-ready images,

<https://jcb.rupress.org/fig-vid-guidelines>.

Thank you for your attention to these final processing requirements. Please revise and format the manuscript and upload materials within 7 days. If you need an extension for whatever reason, please let us know and we can work with you to determine a suitable revision period.

Thank you for this interesting contribution, we look forward to publishing your paper in Journal of Cell Biology.

Sincerely,

Lois Weisman, PhD
Monitoring Editor

Andrea L. Marat, PhD
Deputy Editor

Journal of Cell Biology

Reviewer #1 (Comments to the Authors (Required)):

K. Muneshige and R. Hatakeyama have made an effort to address our concerns with revisions to the text and additional experiments. Specifically, they have done well to further highlight the unique composition of SEs and potentially link SE formation to cell cycle progression. However, several questions still remain, including more convincing evidence concerning the identity of SEs, whether SEs are truly separate from the vacuole membrane as well as the eventual fate of the SEs. The former two has been partially addressed in this paper; however definitive evidence is still lacking.

Other Notes:

-The authors addressed the observation that Ivy1 is located on both endosomes and SEs in Fig S2F and S2G, but it would be prudent to emphasize that Ivy1 is not a specific SE marker in the main text. Especially since, as the authors mentioned, this misconception still persists in the field.

-Does abolishing the SE formation in Atg18 mutants affect the phosphorylation of the substrates of SE localized TORC1? If so, can this not be used as a readout for SE formation in addition to images? This would also further illustrate the specific function of SEs.

-Is Atg18 only transiently associated with SEs during formation or does the colocalization between Atg18 and SEs persist? How much time does it take between SE formation and resolution?

-If SEs are, in part, linked to cell cycle progression, would it not be beneficial to synchronize the yeast cell cycle in order to investigate exactly when these SEs form? I'm aware the authors have stated that this warrants further study in the discussion, however synchronizing the yeast cell cycle might lead to a better yield of SEs in cells, and further aid the investigation.

Reviewer #2 (Comments to the Authors (Required)):

Overall, I am happy to recommend this paper for publication in JCB.

This revised manuscript from Muneshige and Hatakeyama does a good job in addressing some of the issues identified by the reviewers. Importantly, they do a good job underscoring why the signaling endosome and its formation are important for JCB readers. Several of the additional requested data are now provided, which help increase the rigor of the work. In addition, the expanded data showing more transporters localized to the SE are helpful.

> New data in Figure 3B show quantization of SE formation during the time course of ATG18 induction. The experimental details are missing here though but relevant to R1s comment about the timelapse images in Fig 3E perhaps being taken of cells under nutrient stress because they are not growing over the course of 2 hrs. Are the data in Fig 3B done in a similar time-lapse fashion, or are these time points visualized from a shaking and grown culture at different time points?

> One aspect was how SE are identified and quantified (eg an SE vs a vacuolar 'side blob'). The authors state that Vph1 is tracked to look for Vph1-negative/mNG-Tor1-positive puncta. Is this the criteria for Fig. 3B - the figure legend does not indicate this, the methods section does not describe this, and accompanying example micrographs do not include this? Thus, it is still not clear whether there is a formulaic way in which SEs are identified and quantified other than a qualitative look at juxtavacuolar dots. I think this can be improved through explicit documentation in the methods and figure legends.

>The Dip5 localization experiment is appreciated because it could help show that this endosome localized transporter, which does not localize to the vacuole, can populate SEs and signify their endosomal characteristics. The data (S4A) still bear some explanation because the cherry-Gtr1 in that micrograph shows multiple puncta whereas in all other micrographs, cherry-Gtr1 localizes to a single puncta designated as the cells singular SE.

> The authors provide some airyscan images SE (3F) and comment that they reveal vacuolar subdomains housing Tor1 and Vph1, even though in regular micrographs, their Pearson's is near 1 (see rebuttal letter). I would urge caution in this interpretation because with the low laser settings apparently used here, there is a very stochastic excitation of fluors, which can cause spatially separate populations of red and green emitting proteins. One would need to show coincidences between GFP and mCherry tagged Vph1 and between GFP and mCherry tagged Tor1 to really make this point.

Reviewer #3 (Comments to the Authors (Required)):

The authors have addressed my main concerns with a wealth of new data and validation of tools employed. They have made a more than good faith effort to quantify their key findings and their new data more rigorously support the conclusions of their study. This is a highly interesting topic and the identification of signaling endosomes as a hybrid compartment with contributions from both forward and retrograde trafficking is very intriguing and should be of broad interest to JCB readers.